# Fullerene-decorated PdCo nano-resistor network hydrogen sensors with sub-second response and parts-per-billion detection at room temperature

Tu Anh Ngo [1] ✉, Ashwin T. Magar[1], Minh T. Pham[1], Hoang M. Luong [1,2], Thi Thu Trinh Phan[3], M. Tuan Trinh[3], Michael Jung[4], George K. Larsen [5], Yiping Zhao [1] & Tho D. Nguyen[1] ✉

Hydrogen detection with rapid response and ultra-low detection limits remains a critical challenge for safety and energy applications. Here, we report a fullerene-decorated PdCo nano-resistor network sensor that integrates nanostructuring, alloying, and surface-engineering approaches. The $C_{60}$ layer enhances sensor performance by increasing the surface-to-volume ratio, enabling fast hydrogen diffusion, relieving mechanical stress during cycling, and guiding nanostructure morphology. Our composite device (20 nm $C_{60}$/ 3 nm Teflon AF/5 nm $Pd_{63}Co_{37}$/30 nm Teflon AF) achieves a response time of $0.40 \pm 0.06$ s across 1–100 mbar $H_2$ and detects 40 ppb $H_2$ with a signal-to-noise ratio of 10 at room temperature. A poly(methyl methacrylate) (PMMA) topcoat further improves cycling stability and selectivity under 90% relative humidity and interfering gases. This design provides a scalable approach and opens the door to future adaptation of porous carbon-based frameworks and polymeric interlayers to realize robust, high-performance hydrogen sensors for real-world applications.

Hydrogen plays a central role in clean energy technologies and industrial processes, yet its colorless, odorless nature and flammability require rapid and reliable detection systems[1–4]. The sensors must feature a response time ($t_{90}$) of ≤1 s for $H_2$ concentrations ranging from 0.1 to 10 vol.% (equivalent to $H_2$ partial pressure of 1–100 mbar), a limit of detection (LOD) in the range of 1000 parts per million (ppm) for automotive and 10 parts per billion (ppb) for environmental monitoring applications[5,6].

Recent advancements across both electrical and optical platforms in achieving ultra-fast response times and ppb-level sensitivity in hydrogen sensors have centered on nanostructured Pd hydride systems due to their high selectivity, excellent sensitivity, fast response times, and mechanical and chemical robustness (see Supplementary Table 1)[7–13]. Pd-based alloys (e.g., PdCo, PdAu) dominate high-impact designs. For instance, recent works on optical sensors by Nugroho et al. using scattering spectrum from PdAu nanoparticle@polymer nanocomposite materials[7], and Luong et al. using transmission intensity through PdCo nanocap arrays[10] explicitly demonstrate sub-second response with LODs around single-digit ppm. The LOD could be further improved to a few hundred ppb by making use of the particle swarm optimization algorithm to optimize the sensor array configurations[11], neural-network-based data treatment to correct the

[1]Department of Physics and Astronomy, University of Georgia, Athens, GA, USA. [2]Department of Electrical Engineering, Faculty of Engineering, Chulalongkorn University, Bangkok, Thailand. [3]Department of Chemistry and Biochemistry, Utah State University, Logan, UT, USA. [4]School of Environmental, Civil, Agricultural and Mechanical Engineering, University of Georgia, Athens, GA, USA. [5]Hydrogen Isotope Science Group, Savannah River National Laboratory, Aiken, SC, USA. ✉e-mail: anhngo@uga.edu; ngtho@uga.edu

baseline drift while testing in extreme humidified conditions[14], or by simply stacking the sensors vertically to improve the signal-to-noise ratio (SNR)[12]. Compared to optical-based sensors, electrical transducing method shows great promise for achieving an ultra-sensitive LOD of a few ppb by virtue of the low noise signal readouts. Pham et al. proposed a PdCo composite hole array (CHA) nano-network with a measured LOD of 180 ppb; however, the response time fell short of the 1-s response time benchmark required for automotive applications[9]. Pd-doped oxides (e.g., Pd@$Fe_2O_3$ nanotubes) combine Pd's catalytic activity with oxide heterojunctions, achieving 50 ppb LOD through synergistic surface reactions and p−n junction modulation[15]. Non-Pd materials, such as rGO/ZnO-$SnO_2$ composites, also reach 50 ppb sensitivity via Pd doping and heterostructure engineering, though they lag behind Pd hydrides in speed and require high temperature operation of >300 °C[16]. These strategies highlight the critical role of alloying, nanostructuring, and hybrid material design tailoring for each transducing method and sensing mechanism in pushing the detection speed and sensitivity limits, with Pd-based systems remaining the gold standard for safety-critical applications. To our best knowledge, no single sensor reported to date has simultaneously achieved both ≤1-s response time and a ppb LOD operating at room temperature (Supplementary Fig. 1). Achieving ppb or sub-ppb detection limits not only benchmarks sensor performance under noise-limited conditions requiring high SNR and advanced signal processing[14,17], but also enables critical applications such as trace hydrogen detection in deep-space missions[18], the monitoring of low-abundance hydrogen isotopes like deuterium and tritium in nuclear environments[19], and breath tests for assessing gastrointestinal health[20,21].

In this study, we present a state-of-the-art hydrogen sensor based on a hexagonal nano-resistance network of $Pd_xCo_{100-x}$ alloy fabricated on a fullerene $C_{60}$-coated glass substrate. This design incorporates four synergistic strategies to simultaneously achieve ultra-fast response and ppb sensitivity. First, the PdCo alloy was selected based on prior optimization studies demonstrating faster kinetics than pure Pd or Pd alloys such as PdAu and PdAg under identical structural and testing conditions[10]. In addition, the Co incorporation increases the strain-induced energy barrier to form the hydride effectively suppressing the hysteresis typically seen in pure Pd hydrides[10,22]. Second, sandwiching the $C_{60}$ interlayer between the PdCo and the glass substrate, known to be inert to hydrogen chemisorption and physisorption at low pressures[23,24], offers multiple functional benefits: (i) effectively doubles surface-to-volume ratio (SVR) of the sensing element thanks to the nanoporous structure that facilitates the hydrogen sorption processes through the $C_{60}$ and PdCo interface[25]; (ii) it acts as a mechanical decoupler, enabling stress-free volumetric changes during hydrogenation cycles and enhancing device stability[26,27]; (iii) it serves as a morphological template, guiding the formation of the PdCo nanostructure; and (iv) it provides a chemically and mechanically robust foundation compatible with vacuum deposition[28]. Third, the nanostructuring strategy, combining nanosphere lithography (NSL) and glancing angle co-deposition (GLACD), produces a CHA structure with high SVR. This not only enhances hydrogen sorption kinetics[29] but also amplifies resistance changes through increased electron scattering at grain boundaries and surfaces at the bottle-neck of the sensing elements[9]. In addition, the GLACD process further promotes alloy formation under far-from-equilibrium conditions, yielding defect-rich films with abundant active diffusion pathways[30]. Lastly, surface modification with Teflon AF (TAF) reduces the activation energy for hydrogen sorptions, accelerating response time while maintaining signal strength[7]. Together, these four integrated strategies enable a sensing platform that achieves sub-second response times and ppb-level hydrogen detection under ambient conditions.

Our results in vacuum-mode measurement demonstrate that 20 nm $C_{60}$/5 nm $Pd_{63}Co_{37}$/30 nm TAF CHA sensors with a hole diameter of 450 ± 4 nm and a 500 nm array period achieve a response time of ≤0.8 s over a 1–100 mbar $H_2$ pressure range, with a measured LOD of 144 ppb and an extrapolated LOD of 40 ppb. Remarkably, when the PdCo sensing layer is sandwiched between two TAF layers, the sensor can detect $H_2$ as fast as 0.40 ± 0.06 s within the same pressure range, boasting the measured LOD down to 40 ppb with an exceptional SNR of 10. This $C_{60}$/TAF/$Pd_{63}Co_{37}$/TAF stacked architecture holds promise for detecting hydrogen at single-digit ppb and sub-ppb levels. When coated with Poly(methyl methacrylate) (PMMA), the sensor demonstrates resilience against interference gases such as $CO_2$, $CH_4$, CO, maintains performance at up to 90% relative humidity (RH), and remains robust over hundreds of hydrogen cycling events using $N_2$ as the carrier gas. This low-cost, lightweight, and energy-efficient platform offers a viable path toward meeting key performance metrics for automotive safety, concentration control, and environmental monitoring. These findings also highlight the broader potential of employing interlayers such as carbon-based materials, metal-organic frameworks and polymers with tunable gas permeability for designing next-generation hydrogen sensors.

## Results

The fabrication scheme of a PdCo CHA on fullerene $C_{60}$-decorated glass substrate is depicted in Fig. 1a[9,31,32], and is described in detail in the "Methods" section. First, a sacrificial nanosphere mask was prepared, and this involved depositing a monolayer of 500 ± 10 nm polystyrene (PS) nanospheres onto the substrates in a hexagonal close-packed arrangement. Subsequently, the size of the nanospheres in the monolayer was reduced via an oxygen plasma reactive ion etching (RIE) process by controlling the etching time $t_{RIE}$. These substrates were then used as templates for $C_{60}$ and PdCo alloy depositions in the later steps, and the resultant hole size of the CHA was determined by $t_{RIE}$. Regarding the hydrogen sensing layer, we investigated a series of PdCo alloy compositions with palladium content ranging from 80% to 55%. Among these, the $Pd_{63}Co_{37}$ composition offered the best trade-off between response time and sensitivity[12], and is therefore the primary focus of this study. A detailed discussion of this compositional optimization is provided in the later part of the article. The thicknesses of $C_{60}$ and PdCo are specified in each case with the corresponding atomic force microscope (AFM) images. For brevity, we use the notations PdCo $CHA_{t_{RIE}}$ and $C_{60}$/PdCo $CHA_{t_{RIE}}$ to refer to $Pd_{63}Co_{37}$ and $C_{60}$/$Pd_{63}Co_{37}$ CHAs, with $t_{RIE}$ representing the RIE etching time in seconds. Sensors coated with TAF (and PMMA) are labeled as $C_{60}$/PdCo/TAF(/PMMA) $CHA_{t_{RIE}}$. The nano-architecture of the sensor was verified by a scanning electron microscopy (SEM) image of one of the devices (20 nm $C_{60}$/5 nm PdCo $CHA_{450}$) (Fig. 1b) and its atomic composition $Pd_{63}Co_{37}$ was confirmed by energy-dispersive spectroscopy (EDS) elemental mapping and line scans (Fig. 1c and Supplementary Fig. 2). The measured weight percentages of C, Pd, and Co were 79.5%, 15.4%, and 5.1%, respectively, corresponding to an atomic ratio of 63:37 between Pd and Co (see conversion Supplementary Table 3). The detailed experimental setups for the hydrogen sensing characterization were described in "Methods" and Supplementary Note 3. To mitigate contact and wire resistances, a four-point probe measurement technique was used throughout the paper (Fig. 1a).

To investigate the role of the buffer layer $C_{60}$, we initially assessed the sensing performance of 15 nm PdCo $CHA_{450}$ with and without 50 nm $C_{60}$ underneath (Fig. 2). Both sensors show linear current-voltage (I-V) characteristics, in which the measured resistances (R) of the sensors with and without $C_{60}$ in vacuum are 130.8 Ω and 85.3 Ω, respectively (Fig. 2a). The difference in R can be attributed to slight morphology variations in the bottle-neck regions of the two CHAs (Supplementary Fig. 7) and the interface differences between the substrate (glass or $C_{60}$) and the PdCo layer. Figure 2b shows the sensors' response to a step-wise $H_2$ pulse of 10 mbar. Here, the percentage change in R or sensitivity of the electrical sensor as a function of $P_{H_2}$ is

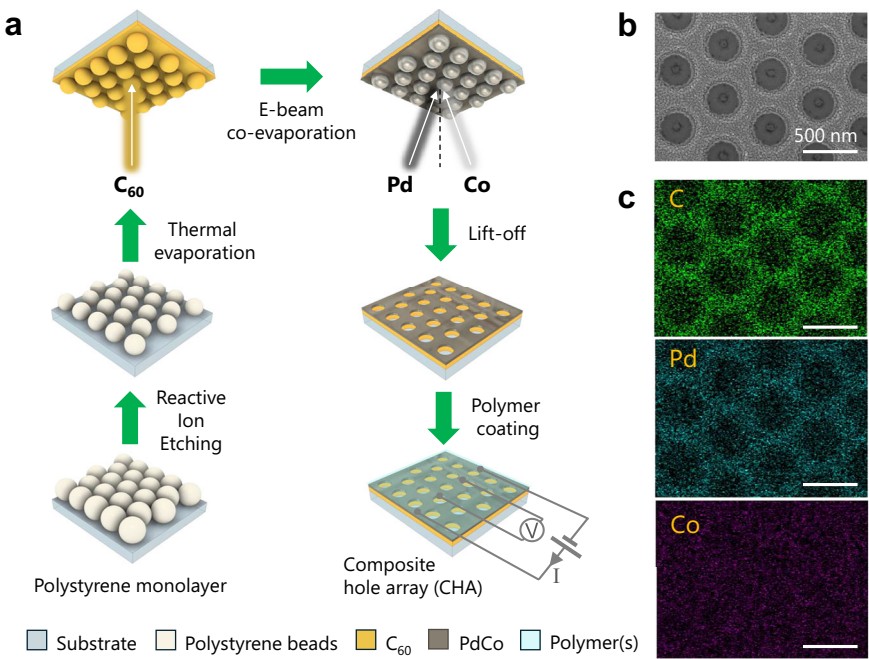

**Fig. 1 | Fabrication scheme and morphology characterization. a** Fabrication process of composite hole arrays (CHA). **b** A typical top-view SEM image and **c** EDS elemental maps of 20 nm $C_{60}$/5 nm PdCo $CHA_{450}$ based on weight percentages. The scale bars correspond to 500 nm. All microscopy images were obtained in triplicate ($N = 3$) and display similar results.

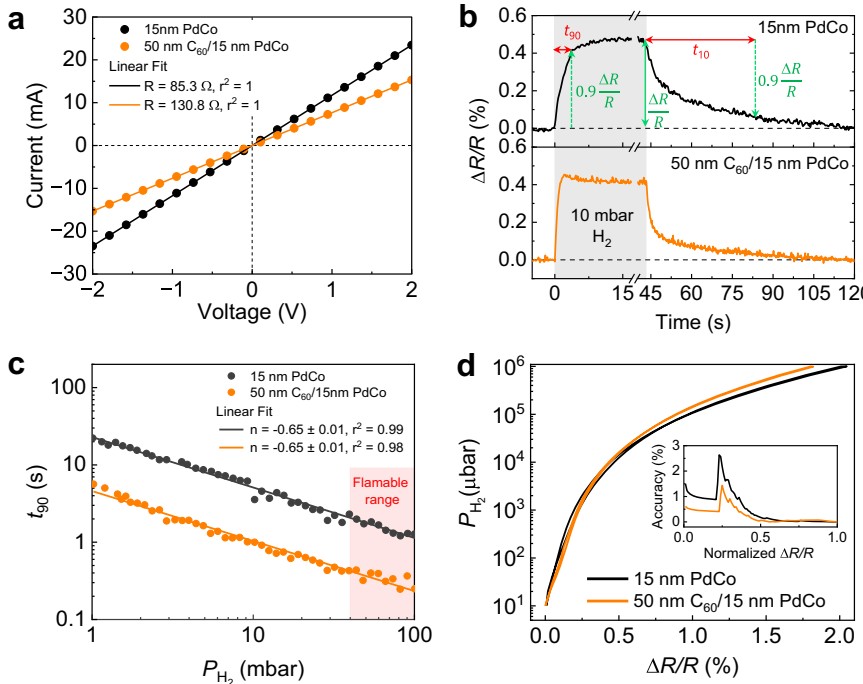

**Fig. 2 | Sensing performances of 15 nm PdCo composite hole array $CHA_{450}$ sensors with and without 50-nm $C_{60}$ buffer layer. a** Current-voltage ($I-V$) characteristics and resistances of the CHAs. **b** Sensors' electrical resistance in response to a stepwise $H_2$ pulse of ~10 mbar (shaded areas). **c** Absorption time $t_{90}$ of the sensors at $P_{H_2} = 1-100$ mbar. **d** Electrical hydrogen sorption isotherms and sensor's accuracy (inset) over $P_{H_2}$ of $10^1$ to $10^6$ µbar. All measurements were performed in vacuum mode at room temperature. Source data are provided as a Source Data file.

defined as:

$$\frac{\Delta R}{R}(\%) = \frac{R(H_2) - R(0)}{R(0)}(\%) \qquad (1)$$

where $R(H_2)$ and $R(0)$ are resistances of the device with and without the presence of $H_2$ gas, respectively. The hydrogen uptake of pure $C_{60}$ at low $H_2$ pressure and room temperature is negligible[23], indicating that the change of $R$ during (de)hydrogenation is primarily due to hydrogen sorption by Pd (Fig. 2b). The positive $\Delta R/R$ observed in both sensors

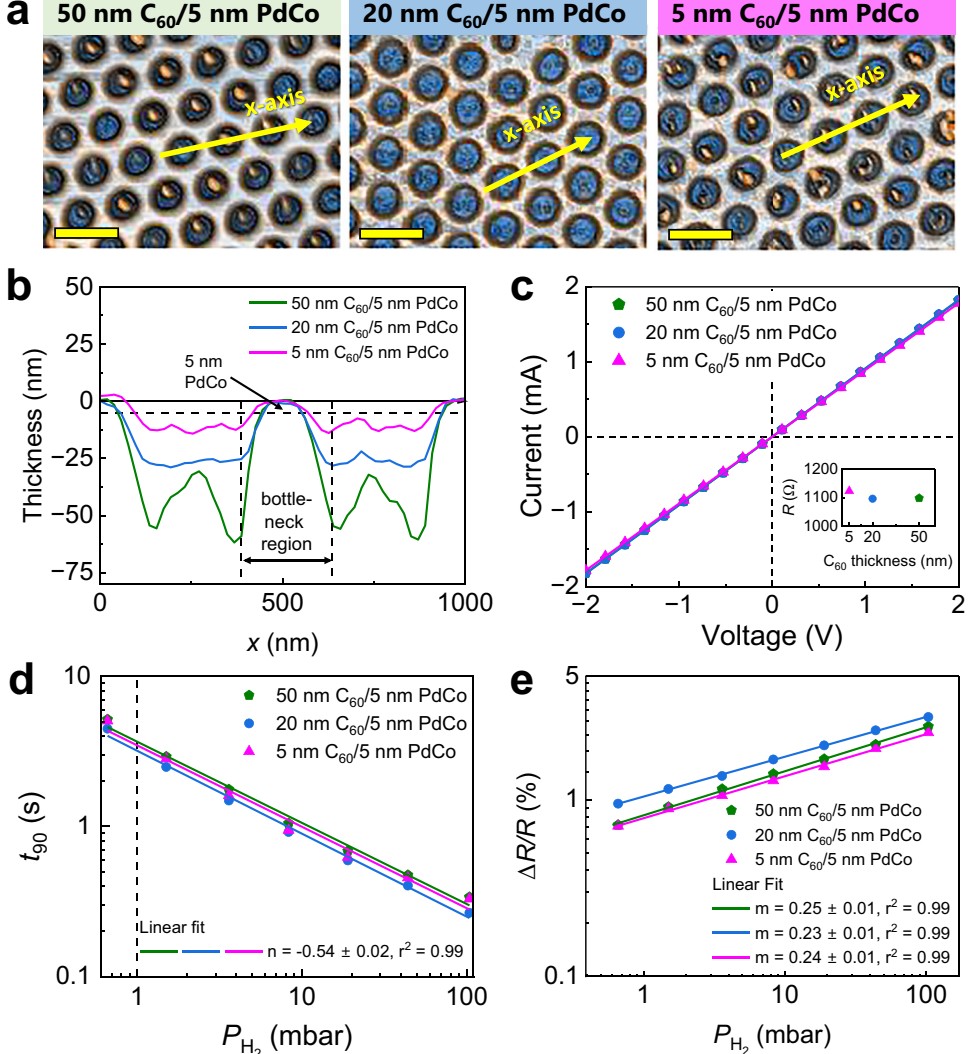

**Fig. 3 | Morphology and sensing performances of $C_{60}$/5 nm PdCo composite hole array CHA$_{450}$ sensors with different $C_{60}$ thicknesses. a** AFM images of CHA$_{450}$ sensors. The scale bars correspond to 500 nm. All microscopy images were obtained in triplicate ($N$ = 3) and display similar results. **b** The line profiles of the CHAs across bottleneck regions (along the $x$-axis denoted in figure a). The profiles are shifted in such a way that the top surface of the CHA is zero. **c** Current-voltage ($I$–$V$) characteristic of the sensors measured in vacuum and calculated resistances (inset). **d** Extracted absorption time $t_{90}$ and **e** sensitivity $\Delta R/R$ of the sensors at $P_{H_2}$ = 0.7–105 mbar. All measurements were performed in vacuum mode at room temperature. Source data are provided as a Source Data file.

indicates that electron scattering dominates the resistance change mechanism, as compared to hydrogen-induced lattice expansion mechanism during hydrogenation, which typically reduces sensor resistance[33,34]. In this case, more scattering centers from the absorbed hydrogen atoms at the interstitial sites were introduced to the system, causing the sensors' resistance to increase. To quantify detection speed, absorption time ($t_{90}$) and desorption time ($t_{10}$) of the sensors at varying $P_{H_2}$ are extracted from resistance dynamics as denoted in Fig. 2b. The commonly used $t_{90}$ for the electrical sensors is defined as the elapsed time from the baseline resistance to 90% of the $\Delta R/R$ saturation, and $t_{10}$ is defined as the elapsed time from the saturated level down to 10% of $\Delta R/R$. The sorption kinetics of $C_{60}$/PdCo CHA$_{450}$ are noticeably accelerated as the $t_{90}$ of the $C_{60}$/PdCo sensor reduce by a factor of >4 over a pressure range of 1–100 mbar compared with those of PdCo sensor (Fig. 2c). In particular, $t_{90}$ of $C_{60}$/PdCo is only 5.6 s at $P_{H_2}$ = 1 mbar, versus 24.7 s of PdCo. Remarkably, it takes ≤1 s for $C_{60}$/PdCo CHA$_{450}$ to detect $P_{H_2}$ ≥ 10 mbar, which is lower than the lower limit of flammable $H_2$ concentration in air (4 vol.% or 40 mbar). Additionally, the time taken for $C_{60}$/PdCo sensor to release $H_2$ across the same pressure range is more than twice as fast as that of the PdCo

sensor (Supplementary Fig. 7). The acceleration in the sensor's response time with the $C_{60}$ interlayer may be attributed to: (i) the crystallographic structure of PdCo thin films on different substrates, or (ii) rapid $H_2$ diffusion through the porous $C_{60}$ walls in the CHA architecture, or (iii) a reduction in the activation energy barrier for $H_2$ adsorption at the $C_{60}$/PdCo interface. To gain insight into the contribution of $C_{60}$ and examine the plausibility of each hypothesis, we conducted X-ray Diffraction (XRD) to analyze the crystal structure (Supplementary Fig. 10) and carried out a controlled experiment using PdCo sensors with a $C_{60}$ top-coating (Supplementary Fig. 13). First, PdCo deposited on both glass and $C_{60}$-coated substrates exhibits a similar crystallographic structure, with the dominant Bragg diffraction peak appearing at ~41.6° (Supplementary Fig. 10b), corresponding to the (111) orientation of the face-centered cubic (fcc) PdCo lattice[35]. The negligible differences in the lattice strain and lattice constant (Supplementary Table 4) make the first hypothesis unlikely. For the second hypothesis, a previous study by Samad et al. reported that the porosity of thermal-evaporated $C_{60}$ thin film can exceed 50% when deposited at room temperature[36]. This aligns with the absence of distinct peaks in the XRD spectrum of $C_{60}$ thin film (Supplementary

Fig. 10a), confirming its amorphous nature and lack of long-range crystalline order of both the $C_{60}$ film and the glass substrate[25,37]. Moreover, the nanohole structure opens up several new pathways for $H_2$ access to PdCo sensing layer through the side walls of the $C_{60}$. Any defect sites or metal-fullerene complexes that have micropores larger than the hydrogen molecules would facilitate the $H_2$ permeation[23]. These combined properties allow hydrogen gas to access the PdCo transducer layer not only from the top surface, as in conventional sensor architectures, but also from the bottom interface via the underlying porous $C_{60}$ film illustrated in Supplementary Fig. 8. This dual-sided accessibility increases effective gas exposure to the sensing material. The SVR is effectively doubled, reducing the average hydrogen diffusion path within the PdCo network and substantially improving the sensor's response time compared to the sensor without $C_{60}$ underneath. The third hypothesis is ruled out, as no enhancement in sensing performance was observed when a 20-nm $C_{60}$ layer was deposited on top of a PdCo sensor (Supplementary Fig. 13). Thus, we can conclude that the enhancement of the response time is solely derived from the increase of the sensing element's SVR by introducing a buffer layer $C_{60}$. In Fig. 2c, the pressure-dependent sorption times of the $C_{60}$/PdCo CHA$_{450}$ sensor follow Sievert's power-law, $t_{90} \propto \left(P_{H_2}\right)^n$[38], as confirmed by a near-linear dependence between $\log(t_{90})$ and $\log(P_{H_2})$ with a similar exponent (or the slope of the linear fit) $n = -0.65 \pm 0.01$ as the PdCo CHA$_{450}$, indicating similar sorption behavior or energy barrier for hydrogen adsorption/desorption in the two sensors, consistent with our previous conclusion. Figure 2d illustrates the sorption isotherms which present the sensitivity $\Delta R/R$ of each sensor across five orders of magnitude of $P_{H_2}$, from $10^1$ to $10^6$ μbar, at room temperature. The hysteresis-free sensitivity of PdCo CHA$_{450}$ is maintained in $C_{60}$/PdCo CHA$_{450}$, as confirmed by accuracy calculations (Fig. 2d inset and Supplemental Note 4). All sensors exhibit accurate readings with <3.0% uncertainty over a wide $H_2$ pressure range thanks to the increase of the strain-induced energy barrier caused by the enhanced structural dislocation in the PdCo system[7,10,12].

Next, the dependence of CHA$_{450}$'s sensing response on the thickness of the $C_{60}$ layer was studied (Fig. 3). In this case, 5-nm-nominal thick of Pd$_{63}$Co$_{37}$ was utilized as a sensing layer and was deposited on substrates coated with 5-nm, 20-nm, and 50-nm thick of $C_{60}$. Figure 3a shows typical AFM images of $C_{60}$/5 nm PdCo CHA$_{450}$ sensors etched at $t_{RIE} = 450$ s, resulting in an average hole diameter of $D_{hole} = 350 \pm 5$ nm (extracted from the line profiles of the AFM images, Fig. 3b). It is worth noting that the polystyrene remnants from the PS beads, left at the center of the holes after the lift-off process, did not interfere with the electrical signals of the devices. The baseline resistances of $C_{60}$/5 nm PdCo CHA$_{450}$ sensors were obtained from the linear $I$-$V$ characteristics (Fig. 3c), with an average resistance of $R = 1100 \pm 10$ Ω. The $C_{60}$/5 nm PdCo CHA$_{450}$ sensors showed a slight improvement in response time and sensitivity ($t_{90} < 3.7$ s and $\Delta R/R > 0.8\%$ at $P_{H_2} = 1$ mbar) compared to $C_{60}$/15 nm PdCo CHA sensor, by virtue of the enhancement of reaction kinetics through nanostructuring[29]. Interestingly, CHA$_{450}$ sensors with different $C_{60}$ thicknesses exhibited comparable $t_{90}$ and $t_{10}$ values (Fig. 3d and Supplementary Fig. 12). These findings further support our assertion that hydrogen diffusion through $C_{60}$ is rapid and not rate-limited in our nano hole-array structure (with a few hundred nm of the hydrogen diffusion path in $C_{60}$). It is feasible to locally probe the diffusion rate of the $C_{60}$/PdCo bilayer using optical techniques[39] and extract the response times at different locations on the films such as near the edges or at the center, such analysis is beyond the scope of the present study. Among the three CHA$_{450}$ sensors tested, the 20 nm $C_{60}$/5 nm PdCo CHA$_{450}$ sensor exhibited the highest sensitivity, which is required for achieving a low LOD. As a result, in the following sensor design, we will maintain the nominal thicknesses of $C_{60}$ and PdCo layers at 20 nm and 5 nm, respectively, while varying $t_{RIE}$ to achieve a

smaller feature size and, consequently, a larger SVR – a key factor for ultra-fast sensing responses.

## Hole size-dependent sensing performance

In the combined NSL and GLACD method[10], the nanohole diameters ($D_{hole}$) or the sizes of the sensing network elements, especially the bottle-neck feature sizes, can be controlled by adjusting the etching time $t_{RIE}$ of the PS beads (Fig. 4). Figure 4a depicts the top-view AFM images of CHAs with varying $t_{RIE}$ from 600 s to 160 s. When $t_{RIE} < 160$ s (Supplementary Fig. 14), a nanotriangle array was achieved instead of a nanohole array, resulting in a discontinuous nano-network. Additionally, an in-house GLACD simulation code with MATLAB[40–42] was utilized to model the morphology and thickness of each layer ($C_{60}$ and PdCo) in each CHA$_{t_{RIE}}$ sensor (see Supplementary Note 7), with the results shown in Fig. 4b, c. The cross-sectional view from GLACD simulations in Fig. 4c aligns well with AFM line profiles, confirming that the $C_{60}$ layer is mechanically stable enough to resist penetration by high-energy metal atoms during deposition. Moreover, $C_{60}$ functions as a critical morphological template, effectively shaping the geometry of the resulting PdCo nanostructures. Morphologically, the hole diameter decreases linearly with $t_{RIE}$ (Fig. 4d). At smaller $t_{RIE}$ values (<300 s), where the bead size after etching exceeds 400 nm, the shadow effect becomes more pronounced, resulting in a U-shaped indentation in the middle of the bottle-neck region, leading to a reduction in the total thickness in this region. The simulated cross-sections along and across the bottleneck region of CHA$_{t_{RIE}}$, along with the thicknesses of PdCo layer ($h_{PdCo}$) projected on a flat surface, are presented in Supplementary Fig. 15. In some regions, the estimated $h_{PdCo}$ can be as thin as 2 nm, and the actual $h_{PdCo}$ might be even thinner than the simulation suggests (Fig. 4c). Moreover, the SVR of the PdCo layer is significantly enhanced as $t_{RIE}$ decreases, due to PdCo being deposited in narrower gaps between the PS beads and on the undulating $C_{60}$ surface. The detailed SVR calculations are presented in Supplementary Note 7.2, and all parameters of 20 nm $C_{60}$/5 nm PdCo CHA$_{t_{RIE}}$ are summarized in Supplementary Table 5. In particular, CHA$_{185}$ has an SVR of -0.75 nm$^{-1}$, which is 1.5 times larger than that of CHA$_{450}$ (0.5 nm$^{-1}$).

The changes in morphology significantly influence the resistances and the sensing performance of the devices, as illustrated in Fig. 4e–g. As the hole diameter increases, the feature size of the resistor network decreases, leading to higher resistance. Figure 4d, e shows that the sensor's resistance increases slightly when $D_{hole}$ increases from 310 nm to 410 nm. In this range, the thickness of the network element does not change, as depicted in Fig. 4c. Therefore, the slight increase in resistance is attributed to the reduction in sensor bottleneck width. However, when $D_{hole}$ increases from 410 nm to 460 nm, a U-shaped indentation forms, causing the network's resistance to increase by nearly three orders of magnitude. A closer inspection reveals that the thickness of the triangular regions (the ends of the resistance network element, see Supplementary Fig. 16) does not change with $D_{hole}$, keeping the resistivity nearly constant in those regions. Consequently, the total resistance of the network is dominated by the resistance at the bottleneck, $R_b$, due to the reduction in thickness because of the shadowing effect as seen in Fig. 4c. The response of the electric current during (de)hydrogenation primarily depends on hydrogen sorption kinetics at the bottleneck[9]. The measured baseline $R$ of CHA$_{t_{RIE}}$ is plotted against the bottleneck's cross-sectional area $A_{bottleneck}$ of the PdCo layer (Fig. 4e). Overall, the $R$ of the nano-resistor network significantly increases as the $A_{bottleneck}$ decreases. This nonlinear relationship was observed in several electrical resistance studies of ultra-thin films at a typical thickness of -10 nm[43–46], and could be explained by three thickness-dependent resistivity models: (i) the Fuchs–Sondheimer model for surface and interface scatterings[47], (ii) the Mayadas–Shatzkes model for grain boundary scattering[46], and (iii) the Namba model considering the ratio of surface roughness to the electron mean free path[48]. Given the complexity of the sensors'

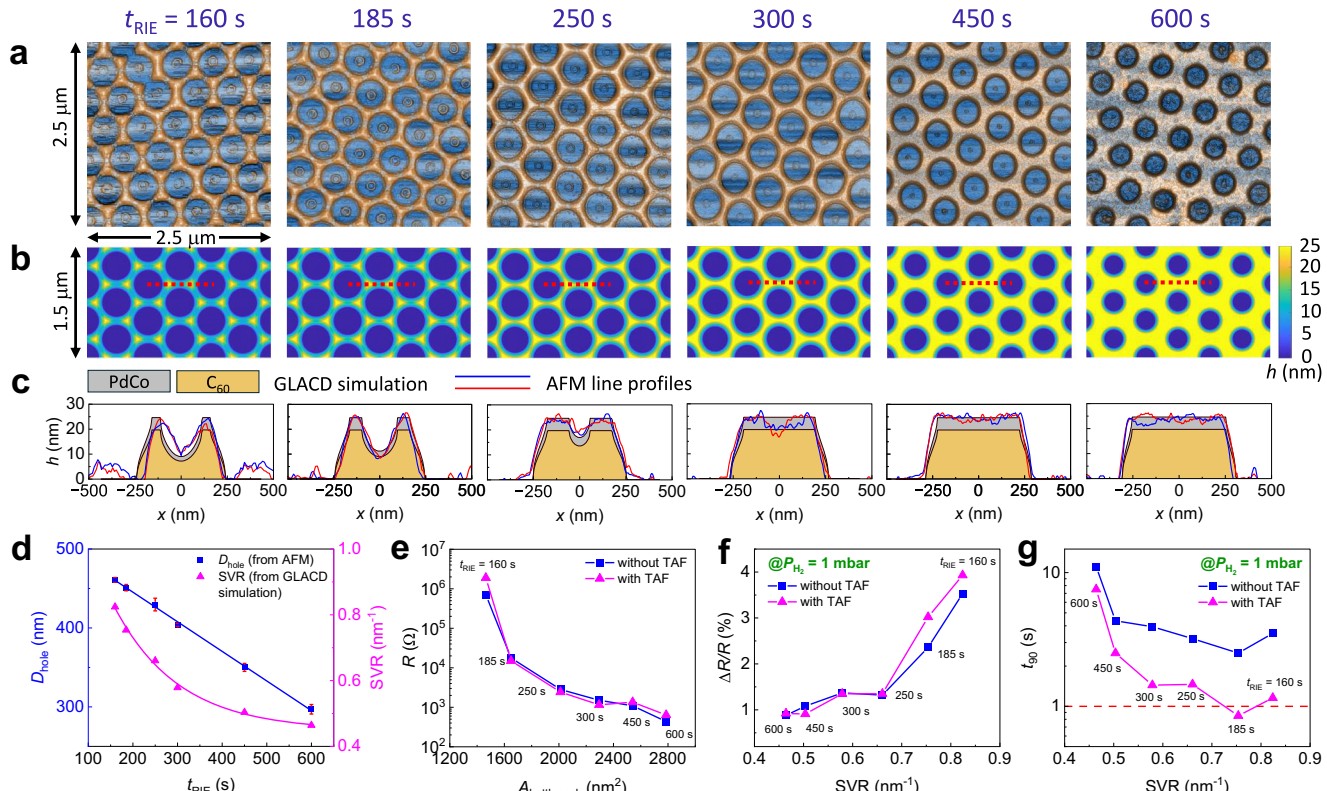

**Fig. 4 | Morphologies and sensing performances of 20 nm C$_{60}$/5 nm PdCo composite hole array CHA$_{t_{RIE}}$. a** Top-view AFM images of CHA$_{t_{RIE}}$ and **b** the corresponding simulated morphologies of CHAs with different etching times. All microscopy images were obtained in triplicate ($N = 3$) and display similar results. **c** Cross-sectional views from glancing angle co-deposition GLACD simulations along the dashed red lines in **b**, showcasing the simulated thickness of each layer, in comparison with AFM line profiles. **d** Estimated hole diameters from AFM averaged from 10 holes (data are presented as mean values +/- SD) and calculated surface-to-volume ratio SVR of PdCo layer from GLACD simulations. **e** Resistances of the 20 nm C$_{60}$/5 nm PdCo CHA$_{t_{RIE}}$ sensors with and without Teflon AF (TAF) coating as a function of the bottleneck cross-sectional area $A_{bottleneck}$. **f** Response time $t_{90}$ and **g** sensitivity of the sensors at $P_{H_2} = 1$ mbar as a function of SVR. Source data are provided as a Source Data file.

structure and the ultra-thin sensing layer, all these scattering mechanisms may be present and influence electron transport. Lacy developed a more general resistivity model, incorporating scatterings from surface, grain boundaries, and surface roughness into a single parameter $\eta$, known as the thickness reduction factor[49]. The theoretical model best fits our experimental data when $\eta = 1.5$ nm for the ultra-thin film regime ($t_{RIE} \leq 300$ s), and $\eta = 0.9$ nm for the uniform thickness regime ($t_{RIE} \geq 300$ s) (see Supplementary Note 7.3 for detailed fitting). The fitting suggests that these scattering effects are more pronounced in the U-shaped region due to the smaller grain boundaries in a thinner PdCo region as shown in SEM images in Supplementary Fig. 16.

All sensors underwent sensing response measurements, with sorption times and sensitivity across $P_{H_2}$ ranging from 100 to 1 mbar being extracted (Supplementary Note 7.4). In order to find the optimal sensor design, $t_{90}$ and $\Delta R/R$ at $P_{H_2} = 1$ mbar of each sensor are depicted in Fig. 4f, g (labeled in blue filled squares). The sensitivity is seen proportional to the SVR of the PdCo layer, as a larger SVR increases the number of available Pd sites for hydrogen adsorption and dissociation[50]. Additionally, response time improves at smaller film thickness, as the hydrogen diffusion path to the interstitial sites of the small PdCo nanocrystalline domains is reduced[29]. Among the sensors, the CHA$_{185}$ sensor exhibits the fastest response time of 2.5 s while maintaining a high sensitivity of ~2.5% at 1 mbar of H$_2$. However, this still falls short of the 1-s response time benchmark required for automotive applications[6]. We explored increasing the Co composition in the alloy and reducing the thickness of the PdCo layer to improve the sensor's response time. Nonetheless, a 5-nm Pd$_{63}$Co$_{37}$ layer emerged as

the best compromise between high sensitivity and rapid response as a higher composition of Co (e.g., 45%) significantly reduces the hydrogen solubility of the sensing element and a thinner nominal deposited thickness (e.g., 3.5 nm) might result in discontinuous resistance network with slower response times (Supplementary Fig. 20).

Perfluorinated polymers, such as Polytetrafluoroethylene (PTFE)[7,51,52] and TAF[12], have been widely used as a coating layer for H$_2$ sensors due to their proven ability to effectively boost the H$_2$ sorption kinetics by lowering the activation energy for both H$_2$ absorption and desorption. Therefore, 30 nm of TAF was then thermally evaporated on all 20 nm C$_{60}$/5 nm PdCo CHA$_{t_{RIE}}$ sensors. Compared to uncoated sensors (labeled in blue filled squares), TAF-coated sensors (labeled in pink filled triangles) display a comparable baseline resistance, a similar rise in $\Delta R/R$, and a reduction in $t_{90}$ with the increase in SVR as shown in Fig. 4e, f. Notably, the TAF coating led to a significant improvement in response time. At $P_{H_2} = 1$ mbar, the $t_{90}$ of each TAF-coated CHA$_{t_{RIE}}$ sensor decreased by ~2 s (Fig. 4g), approaching the 1-s benchmark. In fact, the CHA$_{185}$/TAF sensor surpassed this benchmark, achieving an ultra-fast $t_{90}$ of <0.8 s. Interestingly, in both TAF-coated and uncoated data revealed a kink in the $t_{90}$ curves at $t_{RIE} = 185$ s. Although the CHA$_{160}$ sensor had a higher SVR, its $t_{90}$ was slower than that of CHA$_{185}$ sensor, likely due to the formation of defects, such as vacancies and dislocations, in the few-nanometer-thick film, which could increase the energy barrier for hydride formation[53].

The 20 nm C$_{60}$/5 nm PdCo/30 nm TAF CHA$_{185}$ sensor was found to be optimal and its detailed sensing performance is presented in Fig. 5. In Supplementary Fig. 21a, the normalized sorption kinetics of the sensor in response to $P_{H_2}$ ranging from 100−1 mbar exhibit a superior

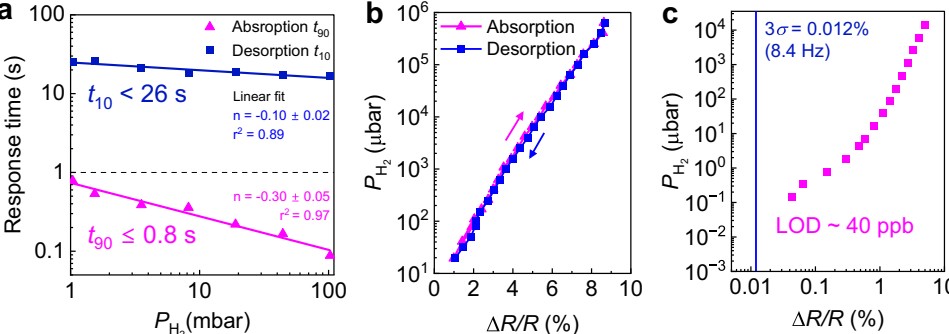

**Fig. 5 | Sensing performances of 20 nm C₆₀/5 nm PdCo/30 nm Teflon AF (TAF) composite hole array CHA₁₈₅.** **a** Absorption time ($t_{90}$) and desorption time ($t_{10}$) of the sensor in response to varying $P_{H_2}$ from 100 to 1 mbar extracted from Supplementary Fig. 21a. **b** Sensor's hydrogen sorption isotherms across five orders of $P_{H_2}$

from $10^1$ to $10^6$ μbar. **c** $\Delta R/R$ plot as a function of $P_{H_2}$ extracted from Supplementary Fig. 21b. The vertical blue line denotes the defined LOD of $3\sigma \approx 0.012\%$ at 8.4 Hz sampling frequency (Supplementary Note 8). All measurements were performed in vacuum mode at room temperature. Source data are provided as a Source Data file.

SNR, with the noise of the acquired signal $\sigma = 0.006\%$ at 12.2 Hz sampling frequency (see Supplementary Note 8), enabling reliable extraction of the response time. Remarkably, the sorption times increase at a much slower rate (with $n_{absorption} \approx -0.30$ and $n_{desorption} \approx -0.10$) as $P_{H_2}$ decreases (Fig. 5a). Particularly, $t_{90}$ remains ≤0.8 s in the 1–100 mbar $P_{H_2}$ range, much faster than any Pd-based electrical hydrogen sensors measured under similar conditions (see Supplementary Table 2). The sensor releases $H_2$ in less than 26 s within the tested $P_{H_2}$ range. In addition, the sorption isotherms of the sensor summarized in Fig. 5b show a hysteresis-free characteristic. To further examine its detection capability, the sensor was tested in diluted $H_2$ gas in an $N_2$ balance (Supplementary Fig. 21b). Although $C_{60}$ is reported to be high permeable to $N_2$[54], the sensor's signal remains unaffected by the $N_2$ pressure variation (see Supplementary Fig. 22a) and the interdiffusion of $N_2$ with $H_2$ has a negligible effect on the sensing performance under vacuum mode measurement within the tested $P_{H_2}$ range as illustrated in Supplementary Fig. 23. Thus, $H_2$ and $N_2$ mixture can be utilized in vacuum mode measurement to reliable extract the sensitivity of the sensor at low partial $H_2$ pressures. The result in Fig. 5c shows that the $C_{60}$/PdCo/TAF CHA₁₈₅ sensor can reliably resolve the hydrogen pressure as low as 144 μbar (≈144 ppb). By defining the LOD as the $P_{H_2}$ at which the detectable resistance change signal = $3\sigma$, where $\sigma = 0.004\%$ at 8.4 Hz sampling frequency (see Supplementary Note 8), extrapolating the sensitivity obtained from Fig. 5c, the TAF-coated CHA₁₈₅ sensor is able to detect the $P_{H_2}$ of 0.04 μbar (≈40 ppb), approaching the DOE requirement of 10 ppb LOD for environmental monitoring applications[5].

Given the significant enhancement in $H_2$ sorption kinetics provided by the TAF top coverage, one might wonder why the $C_{60}$ layer is not entirely replaced by TAF. The fabrication process of the sensor involves the lift-off of PS beads to create the hole array structure. However, TAF does not adhere to the glass substrate as well as $C_{60}$ to survive the lift-off. Despite varying etching times, no conductivity was detected from the TAF/PdCo/TAF CHA sensors. This may stem from the porous nature of the TAF polymer and from the penetration of Pd and Co atoms into the soft TAF layer, both of which can lead to a PdCo thin film with numerous pinholes (see Supplementary Fig. 9) and to discontinuities in the PdCo thin films[55]. Consequently, instead of replacing the entire $C_{60}$ layer with TAF, a very thin layer of TAF was deposited on top of $C_{60}$, followed by PdCo co-deposition and another TAF coating, as depicted in Fig. 6a inset. The characterizations of three sensors with the same stacking structure of 20 nm $C_{60}$/3 nm TAF/5 nm PdCo/30 nm TAF CHA₄₅₀ are shown in Fig. 6. All sensors exhibit linear $I$-$V$ characteristics (Fig. 6a), from which the electric power was extracted and showed in Supplementary Fig. 24. At 0.1 mA of applied source current, the sensor consumes a negligible power of 25 μW or

90 mWh. The average resistance of the sensor on the $C_{60}$/TAF-coated substrate is $R = 2700 \pm 30\ \Omega$, approximately twice that of the sensor on the $C_{60}$-coated substrate. This increase can also be attributed to the slight influence of the TAF's porosity and the penetration of Co and Pd atoms into the ultrathin soft TAF layer, whereas PdCo forms a smoother layer on $C_{60}$ (Supplementary Fig. 9). Here, we demonstrate the hydrogen sensing performance of the CHA₄₅₀-based sensor only because these pinhole effects also prevent the formation of a continuous PdCo nano-network in hole array structures with smaller channels, such as CHA₁₈₅ or CHA₂₅₀ (Fig. 4a). All three tested devices exhibited significantly faster response and release times, with record fast $t_{90} = 0.40 \pm 0.06$ s and $t_{10} = 16 \pm 1$ s across the $H_2$ pressure range of 1–100 mbar as shown in Fig. 6b, the fastest sensor ever reported in the literature at a similar operation conditions (see Supplementary Table 2). The improved sensing response of samples with TAF underneath can be attributed to a combination of reduced grain size observed in the XRD analysis (Supplementary Table 4), efficient gas permeation through the underlayer, and a lowered energy barrier for hydrogen sorption at the TAF/PdCo interfaces[7]. Figure 6c demonstrates the detection capability of one of the $C_{60}$/TAF/PdCo/TAF CHA₄₅₀ sensors through step-wise pressure pulses of diluted $H_2$ in an $N_2$ balance, corresponding to partial $H_2$ pressures ranging from 54.8 to 0.4 μbar. The sensor effectively detected the lowest $P_{H_2} = 0.11$ μbar (~110 ppb) with an impressive sensitivity of 0.2%. Notably, the signal is clearly distinct from the background noise ($\sigma = 0.004\%$ at $f_{sampling} = 8.4$ Hz) and is not affected by the $N_2$ gas carrier pressure (Supplementary Fig. 22b), which is crucial for precise monitoring. The LOD was then confirmed by repeating the measurements for sensors #2 and #3 (Supplementary Fig. 27). The sensitivity averaged from 3 individual sensors across six orders of hydrogen pressures, as shown in Fig. 6d, suggests a reliable detection of 40 ppb with sensitivity of $0.11 \pm 0.03\%$ and an extrapolated LOD as low as 100 part-per-trillion (ppt) (Supplementary Fig. 27d). Our sensor's exceptional performance ranks it among the fastest and most sensitive hydrogen sensors currently available (Supplementary Note 1).

## Stability and selectivity tests

In addition to response time and the detection limit, the sensor's capability to reliably detect $H_2$ under the influence of aging, toxic gases, humidity and temperature is crucial for practical application. While TAF-coated sensors improve hydrogen sorption kinetics, their poor selectivity leaves them vulnerable to degradation in the presence of toxic gases like CO or extreme humidified condition (see Supplementary Fig. 29)[7,12]. To overcome this limitation, a high $H_2$ permeable poly(methyl methacrylate) (PMMA) has been utilized as an additional protective layer (Fig. 7)[7,9,10,56]. It is worth noting that a 50-nm thick

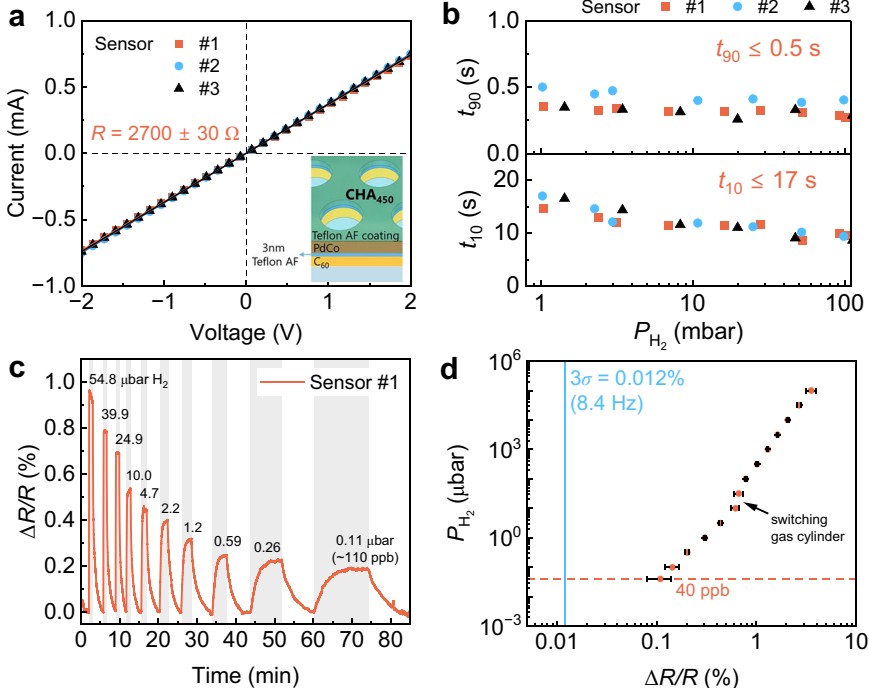

**Fig. 6 | Sensing performances of 20 nm $C_{60}$/3 nm Teflon AF/5 nm PdCo/30 nm Teflon AF composite hole array $CHA_{450}$ sensors. a** Current-voltage ($I$–$V$) characteristics of 3 sensors with the same device structure (inset), **b** absorption time ($t_{90}$) and desorption time ($t_{10}$) of the 3 sensors in 1–100 mbar $H_2$ pressure range. **c** $\Delta R/R$ response of sensor #1 upon partial $H_2$ pressures of 54.8–0.11 µbar measured at 8.4 Hz sampling frequency. **d** Averaged $\Delta R/R$ plot as a function of $P_{H_2}$ with the error bars indicate the standard deviation from 3 devices. Data are presented as mean values +/- SD. The vertical blue line denotes the defined LOD of $3\sigma \approx 0.012\%$ at 8.4 Hz sampling frequency (Supplementary Note 8). All measurements were performed in vacuum mode at room temperature. Source data are provided as a Source Data file.

PMMA coating layer shows negligible effect on the sensors' response times thanks to its high permeability and diffusivity to $H_2$ (Supplementary Fig. 30)[57]. First, we assessed the stability of both a fresh and a 3-month-old (stored in a $N_2$ filled glovebox) 20 nm $C_{60}$/5 nm PdCo/ 30 nm TAF/PMMA $CHA_{450}$ sensor upon 100 cycles of loading/ unloading 2% $H_2$ in $N_2$ using a flow mode set-up (see Methods and Supplementary Note 3). Overall, the sensitivity of the sensors remained stable throughout the cycling test, with a small variation of <5% (Fig. 7a, b). The 3-month-old sensor showed a comparable (de)absorption time to the fresh one (Fig. 7a inset), demonstrating resilience against mechanical stress and strain from repeated lattice expansion and contraction. It can be ascribed to the high degree of rotational disorder of the $C_{60}$ molecules around one of their central axes at room temperature[27,58], which allows the PdCo film to expand and contract with minimal mechanical constraint. Even if chemical bonding occurs at the PdCo/$C_{60}$ interface (see X-Ray Photoelectron Spectroscopy (XPS) analysis in Supplementary Fig. 11), this molecular mobility ensures that the sensor retains mechanical flexibility and stability, mitigating the degradation due to the buckle or global detachment of the PdCo film from the substrate in clamped films[59–61].

Next, we conducted tests to evaluate the sensor resistance to poisonous gases and humidity. In deactivation tests, the sensor was exposed to 5% of $CO_2$, 5% of $CH_4$, and 0.2% of CO, each mixed with 2% $H_2$ in $N_2$, respectively. As shown in Fig. 7c, d, the normalized sensitivity remained within ±4%, well below the ±5% accuracy required for hydrogen sensors[6]. The response time remained almost unchanged (within 1 s) before and after exposure to 5% $CO_2$ or 5% $CH_4$, though a larger fluctuation of ~3 s in absorption time was observed in high concentrations of CO (0.2%) (Supplementary Fig. 31). It is important to note that 0.2% of CO represents an extreme testing condition, as typical urban CO levels are around 10 ppm (~0.001%)[62]. We also performed humidity tests with gas carriers at 40% and 90% relative humidity (RH) compared to the dry condition (Fig. 7e, f). The sensor's sensitivity gradually decreases after each cycle due to water vapor condensation on the sensing elements, which can hinder hydrogen adsorption and dissociation[63,64]. $\Delta R/R$ at 40% RH appears to converge with the signal at 90% RH, suggesting that the effect of humidity may saturate once a certain amount of water is deposited on the sensor surface. The average signal amplitudes in 40% RH and 90% RH were $91 \pm 3\%$ and $87 \pm 1\%$, respectively, compared to the dry reference environment. Notably, the impact of humidity is temporary since the water molecules are weakly bound to the sensor surface and can be cleared by injecting dry air[65]. The sensor's performance recovered after a few pulses of dry 2% $H_2$ in $N_2$ (Supplementary Fig. 32). This result places the $C_{60}$/PdCo/TAF/PMMA CHA sensor among the top-performing hydrogen sensors under extremely humidified conditions (RH = 90%)[8,14]. While this study focuses on idealized conditions, we acknowledge that evaluating sensor performance in air is essential for real-world applications. Some preliminary results with polyvinyl alcohol (PVOH)-coated sensors show that oxygen interference can be mitigated; however, the coating introduces trade-offs in hydrogen diffusion and limits applicability under humid conditions[11,66]. Further optimization of coating thickness and tandem architecture to balance selectivity, response time, and environmental robustness remains an important future direction beyond the scope of this work.

To address the temperature stability of the system and the temperature dependence of resistance, we evaluated sensor performance of one of our CHAs over a range from room temperature to 115 °C and consistently observed hydrogen response (see Supplementary Fig. 33). However, two notable effects emerged at elevated temperatures: (i) increased baseline drift, which may result from thermally induced morphological changes in the nanoparticle network such as expansion or coalescence that alter percolation pathways, and (ii) a significant drop in sensitivity, consistent with suppressed β-phase hydride

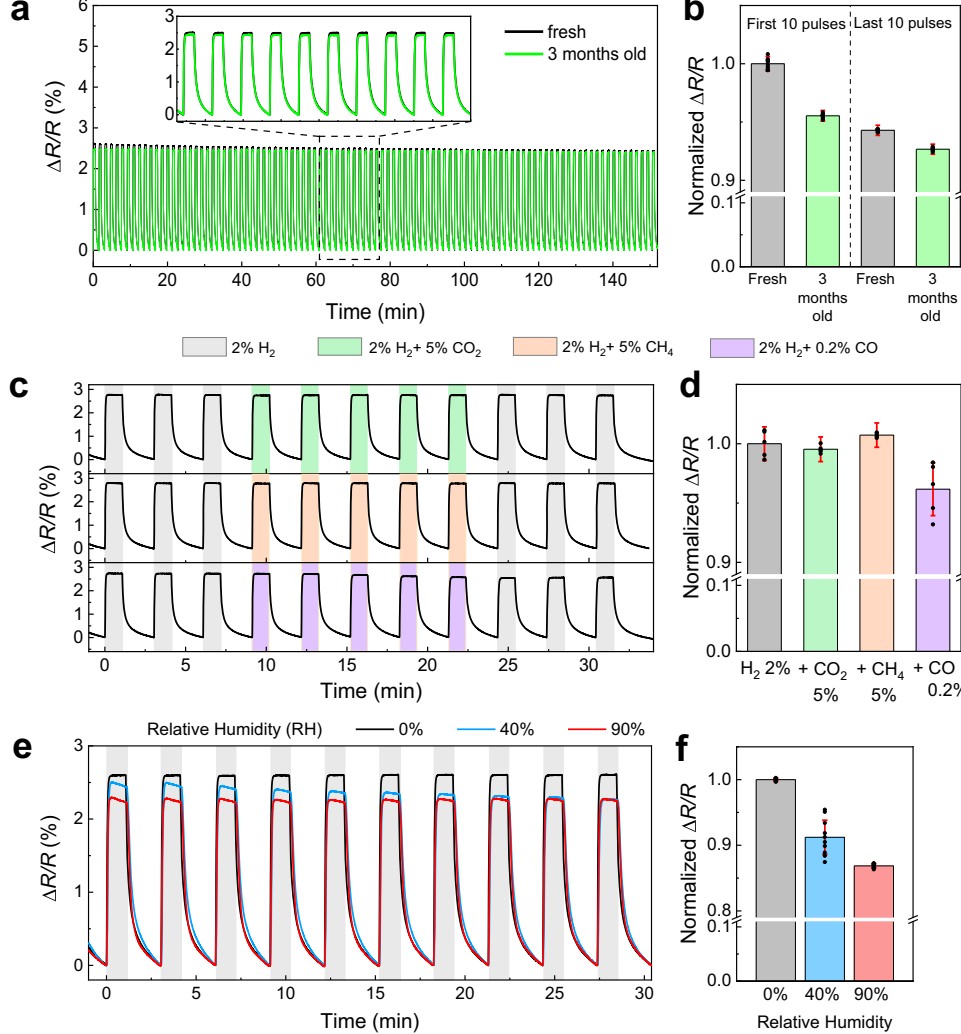

**Fig. 7 | Aging and selectivity tests of 20 nm C$_{60}$/5 nm PdCo/30 nm Teflon AF/ PMMA composite hole array CHA$_{450}$ sensors. a** $\Delta R/R$ responses of fresh and 3-month-old samples upon 100 cycles (30 s/60 s of loading/unloading) of 2% H$_2$ balanced in N$_2$ with a total flow of 400 ml/min. **b** Normalized sensor signal of the first and last 10 pulses in figure a. The error bars denote the standard deviation from 10 cycles. **c** Time-resolved $\Delta R/R$ response of the sensor to 3 pulses of 2% H$_2$ followed by 5 pulses of 2% H$_2$ + 5% CO$_2$, 2% H$_2$ + 5% CH$_4$, 2% H$_2$ + 0.2% CO; then followed by 3 pulses of 2% H$_2$. **d** Normalized sensor signal to the one obtained with 2% H$_2$ in N$_2$ flow. The error bars denote the standard deviation from 5 cycles. **e** Time-resolved $\Delta R/R$ response of the sensor to 10 pulses of 2% H$_2$ with different relative humidity (RH) and **f** normalized signal to the one obtained with 2% H$_2$ in dry condition. The error bars denote the standard deviation from 10 cycles. Data in Fig. 7b, d, f are presented as mean values +/- SD. All measurements were performed at room temperature, using N$_2$ as carrier gas. Source data are provided as a Source Data file.

formation and enhanced hydrogen desorption kinetics[8]. These findings indicate that room temperature offers the best balance between baseline stability and sensing performance for our PdCo nano-network structure.

## Discussion

In summary, we demonstrate a PdCo nano-resistor hydrogen sensor enhanced with a fullerene C$_{60}$ interlayer, which combines nanostructuring, alloying, and surface/interface engineering strategies. The incorporation of C$_{60}$ improves performance by increasing the active surface area of the sensing element, facilitating hydrogen transport, mitigating stress during cycling, and directing the nanostructure formation. This unique and high-throughput design simultaneously achieves ultra-fast and ultra-low H$_2$ detection. In one CHA$_{185}$ configuration ($D_{hole}$ = 450 ± 4 nm) consisting of a stacking 20 nm C$_{60}$/5 nm Pd$_{63}$Co$_{37}$/30 nm TAF, the sensor demonstrates a response time of ≤0.8 s over a hydrogen pressure range of 1–100 mbar, achieving a LOD of ~144 ppb. In another CHA$_{450}$ configuration ($D_{hole}$ = 350 ± 5 nm) with TAF on both sides the sensing element PdCo (20 nm C$_{60}$/3 nm TAF/

5 nm Pd$_{63}$Co$_{37}$/30 nm TAF), the sensor can detect H$_2$ as fast as 0.40 ± 0.06 s within the same pressure range, boasting an exceptional measured LOD down to 40 ppb level. The addition of a PMMA coating provides excellent selectivity against interfering gases (CO$_2$, CH$_4$, CO) and ensures stable performance under up to 90% relative humidity with N$_2$ as a gas carrier. The sensor also maintains high sensitivity after extensive cycling and prolonged storage under inert conditions. It is noteworthy that the high-performance sensors were fabricated using simple, scalable methods that are inherently compatible with standard semiconductor processes, making the design a strong candidate for industrial integration. A detailed discussion of the fabrication steps, scalability, and potential challenges related to large-area uniformity, material cost, integration, and long-term stability is provided in Supplementary Note 10. Our CHA architecture with an interfacial layer of C$_{60}$ and TAF demonstrates strong potential to satisfy the demanding performance and stability requirements of hydrogen sensing in both automotive and environmental monitoring applications. More broadly, this materials design framework, which emphasizes scalable, permeable, and stable interlayers, can guide future strategies for

engineering surface and interface layers (e.g., using carbon-based[67], metal–organic frameworks[68] and polymers[61]) across diverse chemical sensing platforms.

## Methods

### Materials

Polystyrene (PS) nanospheres (Polysciences Inc., average bead diameter $D = 500 \pm 10$ nm, CV = 2%) and ethanol (Sigma-Aldrich, 98%) were used to fabricate the nanosphere monolayers. Sulfuric Acid (Lab Alley, 93%) and hydrogen peroxide (Lab Alley, 30%) were utilized for the petri dish and substrate cleaning processes. Polymethyl methacrylate (PMMA), fullerene $C_{60}$ (99.9%), and acetone (≥99.5%) were purchased from Sigma-Aldrich. Teflon AF 2400 (TAF) was obtained from Dupont. Palladium (99.95%), and cobalt (99.95%) from Kurt. J Lesker Company were utilized for electron beam depositions. Deionized (DI) water (18 MΩ.cm) was used for all experiments.

### Composite nanohole array (CHA) fabrication

The composite nanohole array (CHA) fabrication started with the growing of hexagonal close-packed nanosphere monolayers on glass substrates or silicon wafers using the air/water interface method[31,69,70]. First, a 14-cm diameter petri dish was cleaned with piranha solution (4:1 mixture of sulfuric acid and hydrogen peroxide) and subsequently filled with 22 ml of DI water. Concurrently, a separate colloidal suspension was prepared by mixing 300 μl of the stock PS nanosphere solution, 1000 μl of DI water, and 650 μl of ethanol. This suspension was continuously dispensed via a syringe pump (kdScientific, series 100) at 0.012 mL/min onto the water-filled petri dish. The nanospheres were allowed to self-assemble for ~3 h until the monolayer formed large domains covering ~75% of the water surface. Substrates were then carefully slid horizontally beneath the floating monolayer. Finally, excess water was drained and the coated substrates were then left to dry overnight. The nanosphere diameter $D = 500$ nm was chosen since the monolayers can have the highest quality with this size. The PS monolayer, then, went under the RIE process using Trion Phantom III RIE Etcher (40 mTorr $O_2$, ICP power = 25 W, RF power = 10 W). The etching times were adjusted from 600 s to 150 s for the bead size reduction. In the next step, fullerene $C_{60}$ was thermally evaporated on top of the templates with the deposition rate of 0.26 Å/s, followed by electron beam co-deposition of Pd and Co with 0.05-nm/s total deposition rate yielding a thin $Pd_{63}Co_{37}$ alloy film. To increase the mixing uniformity between Pd and Co, the sample holder was rotated azimuthally with a constant rotation rate of 80 rpm during the deposition process. The substrate was then cut into a standard size of $10 \times 5$ mm$^2$ followed by monolayer lift-off by using the scotch tape technique. In some cases, the device was coated by a layer of TAF and a PMMA film. For TAF coating, the TAF powder was thermally evaporated at the rate of 0.02 nm/s to form a uniform coating with a nominal thickness of ~30 nm[12]. For PMMA coating, PMMA powder was mixed in Acetone at 10 mg/mL concentration at 80 °C until fully dissolved. The solution was then cooled down to room temperature before being spin-casted on top of the hole array at a speed of 5000 rpm for 120 s, followed by the final soft bake at 85 °C for 20 min. The approximate thickness of the coating PMMA layer is 50 nm[10].

### Morphology, composition and crystal structure characterization

The nanomorphologies of fabricated samples were measured by an AFM (Park NX10), and the AFM images were then analyzed using XEI - Image Processing and Analysis Software and Gwyddion[71].

SEM was performed with a SU-9000 system, Hitachi (resolution of 0.4 nm at 20 kV). EDS elemental mapping was performed with 150 mm Oxford XMaxN detector.

XRD patterns were measured using a powder diffractometer (XRD, Rigaku) with Cu-Kα radiation (λ = 1.54056 Å, 40 kV, and 40 mA).

The range of 2 theta angle from 39° to 44° with speed 0.1 degree per minute and 0.03° for the sampling width for all thin film samples on the glass substrate.

XPS spectra of thin films on Si (110) wafer were taken using Thermo Scientific Nexsa G2 XPS system with Al Kα radiation (1486.6 eV) and the line width of 0.25–0.30 eV.

### Sensing characterization

The sample was mounted in a home-built $H_2$ gas cell[70]. The resistance of the CHAs was measured by a collinear 4-point probe using a SourceMeter KEITHLEY 2635B as a constant current source. In vacuum mode measurement, $H_2$ gas pressure was monitored by three independent pressure transducers with different ranges which cover the pressure range of 2.7E-6 to 1.1 bar (two PX409-USBH, Omega and a Baratron, MKS). By using diluted 4% or 100 ppm or 10 ppm $H_2$ balanced in $N_2$, we can control the partial $H_2$ pressure as low as 40 μbar (~40 ppm) or 100 nbar (~100 ppb) or 10 nbar (~10 ppb), respectively. In flow mode measurement, 4% $H_2$ in nitrogen mixture gases (Airgas) were diluted with ultra-high purity nitrogen gas from Airgas company to targeted concentrations by a commercial gas blender (GB-103, MCQ Instruments). The gas flow rate was kept constant at 400 ml/min at 1 atm for all measurements. All experiments were performed at room temperature if not stated otherwise.

### Abridged disclaimer

The view expressed herein do not necessarily represent the view of the U.S. Department of Energy or the United States Government.

### Reporting summary

Further information on research design is available in the Nature Portfolio Reporting Summary linked to this article.

## Data availability

Source data are provided with this paper. Additional information can be obtained from the corresponding authors upon request. Source data are provided with this paper.

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

## Acknowledgements

This material is based upon work supported by the U.S. Department of Energy's Office of Energy Efficiency and Renewable Energy (EERE) under the Hydrogen and Fuel Cell Technologies Office (HFTO) and Funding Opportunity in Support of the Hydrogen Shot and a University Research Consortium on Grid Resilience, Award Number DE-EE0010742. T.A.N. acknowledges the Graduate School, the University of Georgia for the Graduate School Research Assistantships during the 2023–2024 fiscal year.

## Author contributions

T.A.N. and T.D.N. designed the experiment. T.A.N., A.T.M., M.T.P., and H.M.L. fabricated the samples, performed the sensing characterizations and analyzed the experimental data. T.A.N. contributed to the theoretical calculation, simulation and analysis. T.T.T.P. and T.T. performed XRD analysis. M.J. performed XPS analysis. T.A.N. wrote the first draft of the paper. T.A.N., T.D.N., Y.Z., G.K.L., and H.M.L. edited the paper. T.D.N. and Y.Z. were responsible for grant acquisition, project planning and group managing. All authors have approved the final version of the manuscript.

## Competing interests

The authors declare no competing interests.
