## [Peer Review File · Nature Communications]

Fullerene-decorated PdCo nano-resistor network hydrogen sensors with sub-second response and parts-per-billion detection at room temperature

Corresponding Author: Ms Tu Anh Ngo

Version 0:

Reviewer comments:

Reviewer #1

(Remarks to the Author)

I have major comments for authors before publication:

1. Clarify the specific contributions of the hexagonal PdCo nano-resistor network design compared to other state-of-the-art sensors. While the paper mentions the improvements in response time and detection limit, explicitly state how this design surpasses existing methods, particularly in robustness and scalability.
2. Include more details about the reproducibility of the sensor's performance across different samples. The manuscript would benefit from presenting error margins or statistical analyses (e.g., standard deviations) for key metrics such as response time (t_{90}) and sensitivity ($\Delta R/R$).
3. Provide a deeper explanation of the role of the C60 buffer layer in enhancing the sensor's performance, particularly in mitigating the clamping effect. Expand on how the interaction between C60 and PdCo improves hydrogen uptake and response time.
4. While the manuscript highlights the sensor's fast response and low detection limits, include a more comprehensive table or figure comparing these metrics to other published hydrogen sensors. This will strengthen the claim of the sensor's superior performance.
5. Address the sensor's long-term performance under varying environmental conditions. While aging and humidity tests are discussed, include additional details about the sensor's stability and potential degradation mechanisms over extended periods or harsher conditions.
6. The manuscript mentions trade-offs between sensitivity, response time, and material composition. Provide a more detailed discussion of these trade-offs, particularly how different PdCo compositions (e.g., Pd₆₃Co₃₇) were optimized for balanced performance.
7. Discuss the scalability of the fabrication process and the feasibility of deploying these sensors in real-world applications. Include potential challenges in mass production and integration with existing technologies.
8. Include additional validation for the results presented. For instance:
More detailed characterization of the nanostructure using advanced techniques like X-ray diffraction (XRD) or Raman spectroscopy to complement the SEM and AFM analyses.
Verification of the sensor's performance using independent measurement techniques to ensure robustness.
9. The addition of Teflon AF and PMMA layers significantly influences sensor performance. Discuss how the thickness and uniformity of these layers impact key metrics and whether these coatings introduce any limitations.
10. Provide a clear section on potential future work, focusing on enhancing selectivity further and addressing interference from other gases or environmental factors.

Reviewer #2

(Remarks to the Author)

First, I would like to congratulate the authors for such an impressive work. I follow closely the work from the group and it is no surprise that they achieve this. The sensor reported accomplishes an unprecedented combination of low limit of detection and high speed. The authors assigned the success to the addition of C60 to PdCo nanohole arrays, a platform that the group develop and work a lot on, that provides extra surfaces yet interact minimally with the active element, allowing it to expand/contract freely during sensing. This conclusion (if true, see my comments below) will inspire other strategies aiming

for similar goal. The reported methods are detailed, and the figure/discussion are clear. I would love to see this published in Nat Com, however, only if the authors can clarify the major questions/suggestions below.

(Apology for my comments below that are random both in terms of importance and order they refer to in the text).

- One missing, critical point in this work (that a Nat Com paper should have) is the general insight that later can be extended to or inspire other fields. Here, author use C60, but no fundamental design/understanding derived. So further characterization/model are required to back up the argument. Also, what other materials may be used to invoke similar effect?
- Related to that note, I find the argument explaining the reasons behind the enhancement to be unjustified and at times contradictory. (i) C60 is thought to increase the SVR. How? Surface here only deals with the material that interacts with H₂, exhibits properties change, which in turn converts into signal in the sensor reading. In fact, in the calculation authors include the surface of C60 but exclude its volume. Both S and V must be evaluated from the same system. Or am I wrong here? Another point for my argument is since authors said C60 does not interact with H₂, it is then as if it wasn't there, which then should not be included in the surface area consideration. (ii) Even if the assumption by authors is correct, how come C60 is connected to SVR even with the thin film configuration (Fig 2). Here the only access for H₂ to PdCo is only through the top (the PdCo surface), since H₂ access through the bottom of PdCo is only through the side walls of C60, which is much smaller than the sample lateral area. (iii) C60 is also thought to increase sensitivity (i.e. through hydrogen loading) due to the minimum clamping between it and the PdCo. However, author also argues that C60 increases the PdCo wettability. These two are contrary to each other. So which one is true? I think authors should study in detail this interface and see if there is any chemical bonding formation. (iv) Further contrasting argument related to point (iii) is that such enhancement effect is also present when there is FTA in between C60 and PdCo. So what is the role of C60 here then? Does FTA also have similar effect as C60? If so, why then it is thickness-dependent? And the thickness dependency also shows rather contradicting results. If it really due to clamping being less, the thicker the FTA, the less the clamping effect is. However, for 50 nm, the thickest FTA studied, the improvement is actually the lowest. All in all there are major investigation and analysis required to reach any conclusion for the study.
- Results and Discussion headers are missing.
- Please discuss recent strategies in improving sensitivity (to ppb level) in H₂ sensors (electric, optic, ect). This will benefit the readers immensely.
- In the real setting, is there a need to measure H₂ down to ppt level?
- In is mentioned that GLACD method enhances quality, morphology, and SVR. How is it so?
- Related to above, it is said the method will get the sample to "their full potential." This is highly speculative and unjustified. Also, what are their full potential? Is there any fundamental ceiling to this?
- Since the LOD is the result of extrapolation, I don't think it is appropriate to use it in the title and abstract, as it may trick readers. Only highlight such a number in these places if it is demonstrated, or explicitly write that it is an "extrapolated LOD"
- Regarding the extrapolation, what is the basis for it? I think it is never mentioned. Or the authors just extend the line following the data?
- Again regarding extrapolation, I don't think it is justified to claim a value from an extrapolation 3 orders of magnitude higher/lower than the last experimental data point (here from 100 ppb to 0.1 ppb). Many things/trends can happen within that expansive range.
- Another main concern is the use of idealized environments throughout the study. Looking at the state of the field now, where it has progressed immensely, I would highly encourage that we start using "real environment" more, especially for a publication in Nat Com. Thus, I am curious to see how the sensors respond in (synthetic) air. This is important in general, but even more for the authors since they claim that their sensors fulfil the strictest requirement for environmental monitoring (which ironically comprises air as background). Achieving what the authors did is impressive, but this does not reflect the real condition at all. In air, there will be many complex reactions being involved. How does the proposed concept perform in air? If not, how far is it from the ideal situation? As such, similar to my comment above, whenever ideal conditions are used, please also have an explicit statement regarding it when mentioning the performance (especially in title and abstract).
- Related to above, in the contamination tests, I encourage authors to instead expose the sensors to constant dose of contaminations, instead of having them in cycle together with the hydrogen. This, again, is closer to the real conditions. I am curious to see the response to humidity using this scenario, as based on my experience, PMMA would absorb water and eventually deteriorates the sensor's performance.

Reviewer #3

(Remarks to the Author)

The authors described how to fabricate unique hydrogen sensing element, a PdCo composite hole array on a fullerene-decorated glass substrate, and evaluated the sensing times and detection limits for H₂ concentrations were evaluated in relation to the structure of the sensing element. By fine-tuning the structure of the PdCo composite hole array, they achieved a reaction time of less than 1 second and a detection limit at ppt level. It is noteworthy that a high-performance sensor element can be constructed using a simple fabrication method, and the structure of this element may also be applicable to gas sensors other than hydrogen. However, I believe that further discussion is needed on the relationship between the structure and performance of sensing elements before publication, the authors need to fully and in detail address the following points:

[1] The authors need to explain the crystal structures of the PdCo and C60 layers, as this information is crucial for understanding the hydrogen absorption properties of the PdCo layer and the H₂ gas behavior through the C60 layer.

[2] The effect of the C60 layer is unclear. There seem to be a contradiction between the following two considerations. In fig.

2, the authors pointed out that the C60 layer mitigated the clamping effect on the PdCo layer, maximizing the hydrogen uptake and the sensitivity. Meanwhile, in fig. 6, the TAF layer inserted between the PdCo and C60 layers was useful to improving sensitivity. If the TAF layer also works to mitigate the clamping effect, the effect of C60 is thought to be only due to its weak adhesion to the substrate.

[3] What about temperature stability in resistance measurement? Since resistance in metals depends on temperature, it is possible that the baseline of resistance $R(0)$ can shift depending on temperature. While the baseline of $\Delta R/R$ shown in figs. S18 and S21 of the supplementally materials shows an approximately straight line, the resistance was not shown. I kindly ask the authors to comment on the stability of the temperature of the system during measurements and on the temperature dependence of the resistance for the fabricated sample.

[4] Why were response times measured in vacuum mode? For the same hydrogen concentration, is the response time the same in vacuum mode and flow mode? We believe that the response time in flow mode will be slower than that in vacuum mode, because the H₂ diffusion time in the carrier gas (N₂) may be the rate-limiting step for detection at low H₂ concentration in mixture gas. Furthermore, when applying samples to hydrogen sensors in atmospheric environments, we generally believe that the response time in flow mode reflects reality.

[5] In fig. 7, is the deterioration speed of the uncovered sample (20nmC60/5nmPdCo/TAF CHA450) to interference gases and humidity faster than that of the PMMA-covered sample? Since the data of the uncovered sample was not shown, the effect of the PMMA coating on interference gases and humidity was not clear in the manuscript. Please show the data of the comparative sample related to fig. 7 in the supplementally materials.

Version 1:

Reviewer comments:

Reviewer #1

(Remarks to the Author)

Authors already addressed properly

Reviewer #2

(Remarks to the Author)

First I would like to thank the Authors for doing an extraordinary work for their revision. As such, I think now this is an exceptional work and therefore I recommend for publication. I believe this will be accepted very well by the community and will be an important piece for the future development. However if I may request one more thing? There are a few valuable discussion in the Response Letter that does not end up in the final manuscript, for example, comment 1.7. I think it will be a "waste" if it just end there, and there is very likely that some readers would have similar question/concern. Maybe authors can somewhat include this in the SI? Anyway, thanks again and congratulations!!

Reviewer #3

(Remarks to the Author)

In their response, the authors have adequately answered the questions, and the manuscript and supplementary materials have been carefully revised. While most of the responses from the authors were well understood, the answers raised additional questions. Therefore, we consider the manuscript acceptable after addressing the following comments.

Can nitrogen molecules penetrate the C60 layer? If the authors claim that measurements in vacuum mode allow evaluation of the intrinsic properties of the sensing material (C60/PdCo/TAF), they should mention the diffusion of N₂ and H₂ gases in C60.

If N₂ molecules can penetrate the C60 layer, the diffusion rate of hydrogen molecules in the C60 layer in flow mode may be slower than that in vacuum mode because of the existence of interdiffusion of N₂ and H₂ in the C60 layer. In other words, the diffusion of hydrogen in the C60 layer may be the rate-limiting step for the sensing material. This behavior is intrinsic to the sensing material and is distinct from sensing system effects such as chamber geometry and flow rate. Therefore, we believe that the intrinsic response speed of the sensing material cannot be evaluated in vacuum mode. On the other hand, if the C60 layer is impermeable to N₂ gas, the C60 could act as an H₂ filter and the intrinsic reaction rate of the sensing material could be measured in vacuum mode.

Version 2:

Reviewer comments:

Reviewer #3

(Remarks to the Author)

I have understood that the existence of nitrogen hardly affected the reaction time as shown in figure R15 and the nitrogen gas diffusion in C60 layer is not important to evaluate the intrinsic material sensitivity. The authors have addressed my question appropriately. I recommend that the revised manuscript is published to nature communications.

Response Letter:

We thank all the reviewers for their time and insightful comments that helped improved the manuscript substantially. Below are our point-by-point responses to reviewers' comments.

To facilitate clarity in our responses, we have labeled each comment from the reviewers as follows: comments from Reviewer 1 are denoted as 1.x (e.g., 1.1, 1.2), those from Reviewer 2 as 2.x, and those from Reviewer 3 as 3.x, with the numbers assigned in ascending order according to the sequence of comments. In cases where comments from different reviewers are similar or related, we may refer back and forth using these notations for cross-referencing.

We have revised our manuscript and supplementary information in light of these comments by explicitly indicating how the corresponding changes within each comment.

Responses to Comments from Reviewer #1

I have major comments for authors before publication:

Authors' response: We are happy to have an opportunity to address the reviewer's critical remarks, important questions, and insightful suggestions.

1.1. Clarify the specific contributions of the hexagonal PdCo nano-resistor network design compared to other state-of-the-art sensors. While the paper mentions the improvements in response time and detection limit, explicitly state how this design surpasses existing methods, particularly in robustness and scalability.

Authors' Response: We thank the reviewer for this thoughtful and constructive suggestion.

Recent advancements in hydrogen sensors (both electrical and optical platforms) have centered on nanostructured Pd hydride systems, optimized through material selection and structural engineering (**Supplementary Figure 1** and **Supplementary Table 1**). Our work builds on this foundation with a synergistic integration of four key design strategies that collectively enable a low detection limit and ultra-fast response time:

(i) Optimized PdCo alloy composition and thickness to improve response time and hysteresis suppression:

First, the PdCo alloy was selected based on our prior study demonstrating faster kinetics than pure Pd and Pd alloys such as PdAu and PdAg under identical structural and testing conditions.¹ In this article, we utilized the PdCo with the stoichiometry of 63:37 and the optimization (including Co% vs. response/sensitivity trade-offs) is systematically detailed in **Comment 1.6**. In addition, an ultra-thin PdCo layer of only a few nm significantly boosts the sensors' response to H₂ (see **Supplementary Figure 20** for thickness calibration). Second, the Co incorporation increases the strain-induced energy barrier to form the hydride effectively suppressing the hysteresis typically seen in pure Pd hydrides.^{1,2} While PdCo historically exhibits reduced hydrogen sensitivity attributed to lower H₂ solubility, we overcame this limitation by enhancing sensitivity through a C₆₀-decorated substrate and nanostructuring (see the following points (ii) and (iii)).

(ii) Hexagonal nanostructuring via NSL and GLACD to improve the sensing element's surface-to-volume ratio:

We combine nanosphere lithography (NSL) with glancing angle co-deposition (GLACD) to create the hexagonal PdCo composite nanohole arrays (CHAs) with a high surface-to-volume (SVR), which is essential for enhancing hydrogen sorption kinetics.^{3,4} The opening in the CHAs provides more pathways for hydrogen diffusion into the sensing layers, in the meantime to make the sensing element even smaller laterally.⁵ Additionally, due to dominant electron scattering at the narrow bottlenecks of the hexagonal network rather than across the entire resistor, the effective sensing area becomes concentrated in these high-sensitivity regions.⁵ The enhanced electron scattering at grain boundaries and surfaces in these bottlenecks leads to greater resistance changes upon hydrogenation, significantly improving sensitivity (see main text **Figure 4** and **Supplementary Note 7**). Notably, these NSL and GLACD approaches maintain low fabrication costs and high throughput with scalability considerations addressed in **Comment 1.7**.

(iii) Surface engineering with C₆₀-coated substrate to improve response time and hydrogen permeability:

The C₆₀ layer enhances the SVR and overall sensor performance in several keyways. First, the C₆₀ amorphous film, being a robust carbon-based material with internal voids, offers relatively high hydrogen permeability and chemical stability.^{6,7} Second, because hydrogen diffuses relatively fast on C₆₀, the effective SVR of the PdCo network nearly doubles, accelerating response time.^{7,8} Third, the C₆₀ layer serves as a mechanical decoupler between the PdCo sensing network and the rigid glass substrate.^{9,10} This allows the PdCo nanostructures to expand and contract freely during hydrogenation and dehydrogenation cycles, reducing mechanical stress and potentially extending sensor lifetime. Our experiments show consistent performance even after 500 hydrogenation cycles (see **Comment 1.5**). Finally, the C₆₀ layer serves as a necessary morphological template, defining the geometry of the PdCo nanostructure (see main text **Figure 4**). Each point is further elaborated in **Comment 1.3**. For these reasons, the use of a C₆₀-decorated substrate is a central innovation of our sensor design and is highlighted in the manuscript.

(iv) Full-interface TAF coatings for kinetic enhancement to improve response time:

A thin TAF polymer coating on the Pd alloy lowers the activation barrier for hydrogen adsorption, further accelerating response time while preserving signal integrity.¹¹

This unique design, in turn, amplifies the resistance change upon hydrogen exposure, contributing to high sensitivity. When paired with the inherently low-noise characteristics of resistive transducing method, our approach yields a markedly improved LOD. The ultra-thin (~few nm) and undulated PdCo resistor elements promote rapid hydrogen adsorption and desorption dynamics, thereby enabling fast response times while preserving signal strength.

We hope this revised explanation more clearly communicates the novelty and effectiveness of our sensor design in advancing hydrogen sensing performance beyond current state-of-the-art technologies.

Change made to the manuscript:

- The Introduction (**page 4, from line 11**) in the manuscript has been revised accordingly.

- References have been added to the manuscript:
- 19 Aziz, M. T., Gill, W. A., Khosa, M. K., Jamil, S. & Janjua, M. R. S. A. Adsorption of molecular hydrogen (H₂) on a fullerene (C₆₀) surface: insights from density functional theory and molecular dynamics simulation. *RSC Advances* **14**, 36546-36556 (2024). <https://doi.org/10.1039/d4ra06171c>
 - 20 Faiman, D. *et al.* Structure and optical properties of C60 thin films. *Thin Solid Films* **295**, 283-286 (1997). [https://doi.org/10.1016/S0040-6090\(96\)09043-8](https://doi.org/10.1016/S0040-6090(96)09043-8)
 - 21 Weckesser, J. *et al.* Binding and ordering of C60 on Pd(110): Investigations at the local and mesoscopic scale. *The Journal of Chemical Physics* **115**, 9001-9009 (2001). <https://doi.org/10.1063/1.1410391>
 - 22 Bozhko, S. I., Walshe, K. & Shvets, I. V. Control of binding of C60 molecules to the substrate by Coulomb blockade. *Scientific Reports* **9** (2019). <https://doi.org/10.1038/s41598-019-52544-4>
 - 23 Mortazavi, B. Structural, electronic, thermal and mechanical properties of C60-based fullerene two-dimensional networks explored by first-principles and machine learning. *Carbon* **213**, 118293 (2023). <https://doi.org/10.1016/j.carbon.2023.118293>
 - 25 Ai, B. & Zhao, Y. Glancing angle deposition meets colloidal lithography: a new evolution in the design of nanostructures. **8**, 1-26 (2019). <https://doi.org/10.1515/nanoph-2018-0105>

1.2. Include more details about the reproducibility of the sensor's performance across different samples. The manuscript would benefit from presenting error margins or statistical analyses (e.g., standard deviations) for key metrics such as response time (t_{90}) and sensitivity ($\Delta R/R$).

Authors' response: We thank the reviewer for the valuable comments. Here, to demonstrate the reproducibility of our sensors, the error margins for response times and sensitivity averaged from 3 individual sensors for each key configuration are shown in **Figure R1**.

Figure R1. **a** Average response time and **b** sensitivity of 20 nm C₆₀/5 nm PdCo/TAF CHA₁₈₅ sensors. **c** Average response time and **d** sensitivity of 20 nm C₆₀/3 nm TAF/5 nm PdCo/TAF CHA₄₅₀ sensors. The error bars indicate the standard deviation from 3 sensors.

Change made to the manuscript:

- **Figure R1** has been added to Supplementary Information as **Supplementary Figure 26**.

1.3. Provide a deeper explanation of the role of the C₆₀ buffer layer in enhancing the sensor's performance, particularly in mitigating the clamping effect. Expand on how the interaction between C₆₀ and PdCo improves hydrogen uptake and response time.

Authors' response: We thank the reviewer for the constructive suggestion. We have briefly discussed the role of C₆₀ in Comment 1.1, the C₆₀ buffer layer serves multiple key functions:

(i) Enhanced hydrogen access:

C₆₀ is a molecule composed of 60 carbon atoms arranged in a soccer ball-like structure, with each molecule measuring approximately 0.7 nm in diameter.¹² In the solid state, C₆₀ can crystallize into a face-centered cubic (FCC) lattice with an effective interstitial gap of about 0.4 nm, which is slightly larger than the size of a hydrogen molecule (~0.3 nm). However, in our study, 20-nm thick C₆₀ film exhibits no X-ray diffraction (XRD) peaks (see Figure R7 in Comment 1.8 for the XRD spectra), confirming its amorphous nature and the absence of long-range crystalline order. As a result, the intermolecular voids are expected to be significantly larger resulting in a high-porosity interlayer C₆₀.⁷ Additionally, the thermally activated rotational motion of C₆₀ molecules at room temperature further enhances hydrogen diffusion throughout the amorphous matrix.⁹ A further benefit arises from the fact that hydrogen does not form strong chemical bonds with C₆₀ under ambient conditions, enabling rapid molecular transport through the layer.¹³ Our result in Supplementary Fig. 13 shows that the performance of sensors without and with a 20-nm C₆₀ top layer is essentially identical, which further supports this point.

Supplementary Figure 13. Sensing performances of 20 nm C₆₀/5 nm PdCo CHA₃₀₀ sensors with and without 20 nm C₆₀ on top. a I-V characteristics, **b** response time t_{90} and **c** sensitivity of the sensors in response to step wise decreasing H₂ pressure from 100 to 1 mbar. All measurements were performed in vacuum mode at room temperature.

These combined properties allow hydrogen gas to access the PdCo transducer layer not only from the top surface, as in conventional sensor architectures, but also from the bottom interface via the underlying amorphous C₆₀ film illustrated in Figure R2 below. This dual-sided accessibility increases effective gas exposure to the sensing material. The SVR is effectively doubled, reducing the average hydrogen diffusion path within the PdCo network and substantially improving the sensor's response time compared to the sensor without C₆₀ underneath (manuscript Figure 2).

Figure R2. Schematic illustration comparing hydrogen absorption and desorption pathways in PdCo thin films on **a** bare glass substrate and **b** C₆₀-coated substrate.

We tested C₆₀ layers with thicknesses ranging from 5 to 50 nm (manuscript **Figure 3**) and observed negligible change in response time or sensitivity among the sensors. These findings further support our assertion that hydrogen diffusion through C₆₀ is rapid and not rate-limited in our nano hole-array structure (with a few hundred nm of the hydrogen diffusion path in C₆₀). It is feasible to locally probe the diffusion rate of the C₆₀/PdCo bilayer using optical techniques¹⁴ and extract the response times at different locations on the films such as near the edges or at the center, nonetheless such analysis is beyond the scope of the present study.

Change made to the manuscript:

- Manuscript (page 10, from line 1) has been revised accordingly.
- Manuscript (page 11, from line 5) has been revised accordingly.
- References have been added to the manuscript:
 - 30 Morgan, C., Schmalbuch, K., García-Sánchez, F., Schneider, C. M. & Meyer, C. Structure and magnetization in CoPd thin films and nanocontacts. *Journal of Magnetism and Magnetic Materials* **325**, 112-116 (2013). [https://doi.org:https://doi.org/10.1016/j.jmmm.2012.07.052](https://doi.org/10.1016/j.jmmm.2012.07.052)
 - 31 Nguyen, T. D., Wang, F., Li, X.-G., Ehrenfreund, E. & Vardeny, Z. V. Spin diffusion in fullerene-based devices: Morphology effect. *Physical Review B* **87** (2013). [https://doi.org:10.1103/physrevb.87.075205](https://doi.org/10.1103/physrevb.87.075205)
 - 32 Samad, B. A., Belanger, É. & Duguay, C. The effect of substrate deposition temperature on the electrical and optical properties of C60 thin films. *Journal of Applied Physics* **132** (2022). [https://doi.org:10.1063/5.0099291](https://doi.org/10.1063/5.0099291)
- **Figure R2** has been added to Supplementary Information as **Supplementary Fig. 8**.

(ii) Mechanical decoupling via molecular rotation:

In thin C₆₀ films, C₆₀ molecules undergo a temperature-dependent transition between static and rotational states, occurring around 259K.⁹ At typical sensing conditions, the molecules are in a freely rotating state, allowing the PdCo film to expand and contract with minimal mechanical constraint. Even if chemical bonding occurs at the PdCo/C₆₀ interface, this molecular mobility ensures that the sensor retains mechanical flexibility and stability and mitigates the clamping effect. Our experiments show consistent performance even after 500 hydrogenation cycles (see **Comment 1.5** and **Figure R3**).

Change made to the manuscript:

- Manuscript (page 23, from line 1) has been revised accordingly.
 - References have been added to the manuscript:
- 22 Bozhko, S. I., Walshe, K. & Shvets, I. V. Control of binding of C₆₀ molecules to the substrate by Coulomb blockade. *Scientific Reports* **9** (2019). <https://doi.org/10.1038/s41598-019-52544-4>
 - 51 Heiney, P. A. *et al.* Orientational ordering transition in solid $\{\mathrm{C}\}_{60}$. *Physical Review Letters* **66**, 2911-2914 (1991). <https://doi.org/10.1103/PhysRevLett.66.2911>
 - 52 Matelon, R. J. *et al.* Substrate effect on the optical response of thin palladium films exposed to hydrogen gas. *Thin Solid Films* **516**, 7797-7801 (2008). [https://doi.org:https://doi.org/10.1016/j.tsf.2008.03.034](https://doi.org/https://doi.org/10.1016/j.tsf.2008.03.034)
 - 53 Miceli, P. F., Zabel, H., Dura, J. A. & Flynn, C. P. Anomalous lattice expansion of metal-hydrogen thin films. *Journal of Materials Research* **6**, 964-968 (1991). <https://doi.org/10.1557/JMR.1991.0964>
 - 54 Verma, N., Delhez, R., van der Pers, N. M., Tichelaar, F. D. & Böttger, A. J. The role of the substrate on the mechanical and thermal stability of Pd thin films during hydrogen (de)sorption. *International Journal of Hydrogen Energy* **46**, 4137-4153 (2021). [https://doi.org:https://doi.org/10.1016/j.ijhydene.2020.10.163](https://doi.org/https://doi.org/10.1016/j.ijhydene.2020.10.163)

(iii) Morphological tuning:

The presence of the C₆₀ buffer layer plays a critical role in enabling the formation of optimized structural bottlenecks in the PdCo film, as discussed in the main text **Figure 4**. These bottlenecks enhance sensing performance by concentrating resistance changes in regions with a high SVR. Such morphology can only be achieved if the buffer layer is sufficiently robust to withstand penetration by high-kinetic-energy metal atoms during deposition. Previous studies have shown that metals such as Co can readily penetrate conventional organic materials under similar deposition conditions.¹⁵ This limited resistance to metal diffusion likely explains why our sensors function reliably only when the thickness of the bottom TAF layer is very thin (~3 nm) in the main text **Figure 6**. In contrast, C₆₀ effectively resists metal intrusion uniquely enabling the formation of the PdCo nanostructures observed in this study. As a result, C₆₀ is an essential material for achieving the desirable morphology that underpins the sensor's enhanced performance.

Change made to the manuscript:

- Manuscript (page 15, line 1) has been revised accordingly.

1.4. While the manuscript highlights the sensor's fast response and low detection limits, include a more comprehensive table or figure comparing these metrics to other published hydrogen sensors. This will strengthen the claim of the sensor's superior performance.

Authors' response: We thank the reviewer for your keen suggestion. We have added a table below, which compares various hydrogen sensing mechanisms, device structures, transducing methods, and sensing metrics for broader context.

Table R1. Ultra-sensitive state-of-the-art H₂ sensors with LOD < 10 ppm reported for the last 10 years.

Ref.	Transducing method	Material and structure	Measurement conditions	t ₉₀ (s) at 0.1%†	t ₁₀ (s) at 0.1%†	Hysteresis-free?	LOD (ppb)	RH test range	T (°C)	Interference gas test
The U.S. DoE's requirements for environmental monitoring applications				<30s (600ppb)	<30s (600ppb)	Yes	10 ppb	0% – 98%	-30 - 80°C	O ₂ , CO, CO ₂ , hydrocarbons
The U.S. DoE's requirements for automotive applications				<1s (0.1%)	<1s (0.1%)	Yes	0.10%	0% – 98%	-30 - 80°C	O ₂ , CO, CO ₂ , hydrocarbons
This work	Electrical	C ₆₀ /PdCo/Teflon AF/PMMA CHA ₁₈₅	Vacuum & flow mode in N ₂	0.8	26	Yes	144 40*	0% - 90%	RT	CO, CO ₂ , CH ₄
This work	Electrical	C ₆₀ /Teflon AF/PdCo/Teflon AF/PMMA CHA ₄₅₀	Vacuum	<0.5	17		40 1*			
¹⁶ Nature Electronics 2025	Electrical	DPP-DTT thin film on Pt electrode	Flow mode in Air	0.84	6.63	N.A.	192	15% - 80%	20 - 120	EtOH, Me ₂ CO, MeOH, Toluene
¹⁷ Scientific Reports 2024	Electrical	Polyaniline PANI hollow nanotube	Flow mode in N ₂	15 (1ppm)	17 (1ppm)	N.A.	1000	N.A.	N.A.	N.A.

¹⁸ ACS Sensors 2024	Electrical	Pd-Doped α -Fe ₂ O ₃ Nanotubes	Flow mode in Air	49 (200 ppm)	533 (200 ppm)	N.A.	50 (300°C)	0% - 90%	300	C ₃ H ₆ O, NH ₃ , NO ₂ , C ₂ H ₆ O, CO, CH ₄ , CO ₂ , SO ₂ , H ₂ S
¹⁹ Nano Energy 2023	Magneto-optical	PS200/Pd67Co33/TAF nanopatches	Vacuum & flow mode in N ₂	0.4	2.8	Yes	1000	0% - 90%	N.A.	CO, CO ₂ , CH ₄ ,
²⁰ Light: Science & Applications 2023	Opto-electrical	Platinum-silicon nanojunctions	Flow mode in Air	10 (1.3 ppm)	N.A.	N.A.	1000*	N.A.	RT	Air
²¹ Microsystems & Nanoengineering 2023	Thermo-electric	Pt NPs@Al ₂ O ₃ on P+/N+ single-crystalline silicon thermopiles	Flow mode in Air	1.9	1.4	N.A.	1000 (120°C)	N.A.	50 - 120	CH ₄ , C ₂ H ₆ , Acetone, Toluene, CO, EtOH
²² Microsystems & Nanoengineering 2022	Electrical	Pd-doped rGO/ZnO-SnO ₂	N.A.	4 (100ppm)	8 (100ppm)	N.A.	50 (380°C)	N.A.	380	HCHO, C ₄ H ₁₀ , C ₇ H ₈ , CO ₂
²³ Nature Communications 2022	Optical	Pd nanoparticles @PMMA	Flow mode in Air	40 mins (250 ppb)	50 mins (250 ppb)	N.A.	250	N.A.	RT	Air
¹ Nature Communications 2021	Optical	PS500/Pd80Co20/TAF nanopatches	Vacuum & flow mode in Air	0.85	N.A.	Yes	2500	0% - 40%	20 - 42	CO, CO ₂ , CH ₄ ,
²⁴ Sensors and Actuators B: Chemical 2021	Electrical	Pd@Ni foam	Flow mode in N ₂	138 (2%)	300 (2%)	N.A.	7	0% - 99.5%	30 - 70	O ₂ , He, Ar, CO ₂ , Co, NH ₃ , CH ₄ , C ₂ H ₄

⁵ ACS Applied Nano Materials 2021	Electrical	PdCo nanohole arrays	Vacuum & flow mode in N2	10.8	20	Yes	180	0% - 90%	RT	Air, CO
²⁵ Carbon 2021	Electrical	Pd NP on Y2O3/CNT	Flow mode in N2	N.A.	N.A.	N.A.	90 ppb (RT) 5 ppb (100°C)	0% - 75%	100	CO, NO2, H2S
¹¹ Nature Materials 2019	Optical	PdAu NP@TAF/PMMA	Vacuum & flow mode in Air	1	5	Yes	1000 (in Ar) 5000 (in Air)	N.A.	30 - 60	CO2, CH4, CO, NO2, Air
²⁶ IEEE 2015	Electrical	Multiwalled Carbon Nanotubes	Vacuum & flow mode in Air	75 (60 ppb)	N.A.	N.A.	400	N.A.	RT	CO, NH3, CH4, H2S, NO2, and acetone
²⁷ Nanosale 2015	Electrical	Pd NP@CPPy	Flow mode: H2 (abs.), 9 N2:1 O2 mixture (des.)	4.5 (20 ppm)	27 (20 ppm)	N.A.	100	N.A.	RT	N.A.
²⁸ Scientific Reports 2015	Electrical	Pd nanoflower/graphene	Flow mode in N2	80 (10 ppm)	N.A.	N.A.	100	N.A.	RT	N.A.

†If not specified, *Extrapolated data

Change made to the manuscript:

- **Table R1** has been added to Supplementary Information as **Supplementary Table 1**.

1.5. Address the sensor's long-term performance under varying environmental conditions. While aging and humidity tests are discussed, include additional details about the sensor's stability and potential degradation mechanisms over extended periods or harsher conditions.

Authors' response: We thank the reviewer for the thoughtful comment. We address long-term performance and degradation across three environmental scenarios:

(i) Inert and cyclic conditions:

We performed over 500 hydrogenation/dehydrogenation cycles in inert environments to assess durability. While the sensor retained functionality, we observed a gradual decrease in sensitivity. This degradation is primarily attributed to grain coarsening of PdCo nanostructures and the accumulation of residual strain from repeated lattice expansion and contraction. Residual strain from cyclic volume changes alters the energy landscape for hydrogen absorption, increasing the activation barrier for future cycles.⁴

Figure R3. $\Delta R/R$ responses of 3-month-old 20 nm C_{60} /3 nm TAF/5 nm PdCo/TAF/PMMA CHA_{450} sample upon 500 cycles (30/60 seconds of loading/unloading) of 2% H_2 balanced in N_2 with a total flow of 400 ml/min.

(ii) Humid and chemically contaminated environments:

As shown in main text **Figure 7**, the PMMA coating partially mitigates interference from water vapor and toxic gases such as CO. However, it does not fully prevent degradation. While the humidity effect is reversible, with performance recovery upon dry-air purging (see **Supplementary Fig. 32**), long-term exposure to CO and CO_2 leads to irreversible performance loss due to strong C=O bonding on the Pd surface.²⁹ These findings align with prior work by Kim *et al.*,²⁹ and are further supported by our XPS C 1s spectrum in **Figure R4**, which shows persistent C=O bonding on PdCo surfaces stored under ambient lab conditions for several months.

Figure R4. XPS spectrum in C 1s region of PdCo thin film.

(iii) Elevated temperatures:

To address the temperature stability of the system and the temperature dependence of resistance, we conducted additional measurements at varying temperatures from room temperature ($\sim 23^\circ\text{C}$) up to 115°C . The temperature was continuously monitored using a thermocouple affixed to the substrate holder inside the gas cell. As shown in **Figure R5a**, the system exhibited excellent thermal stability, with fluctuations remaining below 0.3°C throughout the measurement period.

In **Figure R5b**, we present the resistance response under hydrogen exposure across the tested temperature range. The baseline resistance R_0 values at each temperature are shown in **Figure R5c**. At lower temperatures, R_0 remains highly stable, following an approximately linear increase from $1260\ \Omega$ at room temperature to $\sim 1320\ \Omega$ at 100°C , consistent with typical metallic behavior where increased electron–phonon scattering raises resistance. We also note that the current used for resistance measurement is limited to $0.1\ \text{mA}$, minimizing Joule heating effects. However, at 115°C , R_0 slightly decreases to $\sim 1290\ \Omega$, which we attribute to possible thermally induced morphological changes in the nanoparticle network such as expansion or coalescence that may alter percolation paths.

As illustrated in **Figure R5d**, the sensor’s sensitivity declines at elevated temperatures, consistent with suppressed β -phase hydride formation and enhanced desorption kinetics at higher temperatures.⁴

Overall, our findings suggest that room temperature provides the optimal balance between baseline stability and sensing performance for our PdCo nano network structure. For field applications, we propose integrating a local temperature control element such as a thermoelectric cooler (TEC) into the sensor housing to ensure consistent operation under ambient fluctuations.

Figure R5. **a** Measured temperatures at the sensor position during the tests. **b** Sensing responses of 20 nm C₆₀/5 nm PdCo CHA₄₅₀ to stepwise decreasing H₂ concentrations with N₂ as the gas carrier and **c** extracted baseline R₀ and **d** extracted sensitivity. The error bars in **d** indicate the standard deviation from 3 pulses.

1.6. The manuscript mentions trade-offs between sensitivity, response time, and material composition. Provide a more detailed discussion of these trade-offs, particularly how different PdCo compositions (e.g., Pd₆₃Co₃₇) were optimized for balanced performance.

Authors' response: We thank the reviewer for your on-point comment.

A key objective in hydrogen sensor design, especially for meeting the DOE targets, is to achieve both high sensitivity and fast response/recovery times. However, there is an inherent trade-off. Sensitivity is often associated with the extent of hydrogen uptake and resultant changes in material properties such as electrical resistance in our case. However, materials with high hydrogen solubility and large sensitivity (e.g., pure Pd) tend to exhibit longer response and recovery times due to phase transitions between α - and β -hydride phases.⁴ To overcome this challenge, alloying Pd with a second metal has been widely explored,⁴ and PdCo is one of the candidates that offer a promising balance between sorption kinetics and sensing performance.^{5,19,30,31} On the one hand, incorporating Co introduces defects and lattice distortions that facilitate faster hydrogen diffusion and improve response dynamics. On the other hand, the presence of Co in PdCo lattice increases the strain-induced energy barrier to form the hydride, which suppresses the α -to- β phase transition and shifts it to higher H₂ pressures,² thus reducing overall hydrogen solubility. So now the key question becomes: how much Co is enough to balance this trade-off, where "enough" is defined as achieving sensing metrics closest to the DOE targets (**Table R1**, page R8)?

To illustrate this point, we present our Co concentration optimization in **Figure R6**. It is worth noting that the response time differences among the four selected stoichiometries are small since we have nanostructure and a very thin sensing layer. In contrast, the sensitivity drops drastically as the Co concentration increases from 37 to 45 atomic %. Based on this trade-off, Pd₆₃Co₃₇ offers the "sweet spot" where we can both obtain moderate hydrogen absorption capacity and fast response time. Co% vs. response/sensitivity extracted at 1 mbar and 9 mbar H₂ have shown in **Supplementary Fig. 20**.

Figure R6. Pd:Co composition-dependent sensing performances of 20 nm C₆₀/5 nm PdCo CHA₁₈₅'s in terms of **a** sensitivity $\Delta R/R$ and **b** response time t_{90} in measured H₂ pressure range of 1 – 100 mbar.

Supplementary Figure 20. **a-b** Pd:Co composition-dependent and **c-d** PdCo thickness-dependent sensing performances of 20 nm C₆₀/PdCo CHA₁₈₅.

Change made to the manuscript:

- Manuscript (page 17, from line 12) has been revised accordingly.

1.7. Discuss the scalability of the fabrication process and the feasibility of deploying these sensors in real-world applications. Include potential challenges in mass production and integration with existing technologies.

Authors' response: We sincerely thank the reviewer for the insightful comment. We have addressed the scalability, feasibility, and potential challenges related to the mass production and integration of our PdCo hexagonal nano-network sensors, as outlined below:

The fabrication process comprises four primary steps: (i) self-assembly of a polystyrene (PS) monolayer, (ii) reactive ion etching (RIE) to shape the nanosphere mask, (iii) physical vapor deposition (PVD) via glancing angle co-deposition (GLACD), and (iv) polymer coating via spin coating or thermal evaporation. These techniques are commonly used in micro/nanofabrication, and particularly RIE and PVD are compatible with standard semiconductor processing workflows. We believe our sensor design is inherently scalable and adaptable, making it a strong candidate for transitioning from laboratory-scale fabrication to industrial production. Nevertheless, we acknowledge the following potential challenges for real-world deployment:

- (i) Uniformity over large areas: Achieving uniform PS monolayers and consistent polymer coatings across large areas is a key concern. Several techniques including spin coating, dip coating, and interfacial assembly have been employed to fabricate monolayers of PS nanospheres. Park *et al.* demonstrated the successful fabrication of a large-area PS monolayer (500 nm bead diameter, identical to our study) with nearly 95% coverage on a 5 cm × 5 cm glass substrate using spin coating and appropriate surfactants.³² Lulek *et al.* reported large-area PS monolayer coating of 4- and 6-inch Si wafers using the air–water interface technique.³³ These self-assembly techniques are well-established and can be automated for scalable production. In addition, scalable coating methods such as slot-die coating, inkjet printing, and blade coating offer viable alternatives to spin coating for large-area fabrication.³⁴ Another approach for large scale production is implementing nanoimprint or UV lithography methods to achieve similar nanopatterns.
- (ii) Material cost: Although Pd is an expensive material, the use of ultra-thin (~several nanometers) PdCo films reduces the total material cost to below 1 USD per sensor.
- (iii) Sensor integration: Integration into packaged sensor systems (e.g., with electrical contacts, microheaters, wireless modules, etc.) requires compatibility with existing CMOS or MEMS technologies. In our case, the sensors are compact (1.0 cm × 0.5 cm) and lightweight, making them well-suited for integration into any commercial packaging. Furthermore, temperature control can be achieved locally using thermoelectric coolers (TECs) to mitigate environmental fluctuations and ensure consistent sensor performance.
- (iv) Long-term stability and robustness: Maintaining structural and functional stability under real-world environmental conditions, such as variations in humidity, temperature cycling, and exposure to contaminants, is an important consideration. We are currently conducting ongoing research on various polymer encapsulation strategies,⁴ local heating,³⁵ and thermal refreshing technique²⁹ to enhance sensor longevity under such conditions.

This combination of high performance, low material cost, and scalable fabrication makes our hydrogen sensor highly promising for future commercialization.

1.8. Include additional validation for the results presented. For instance: More detailed characterization of the nanostructure using advanced techniques like X-ray diffraction (XRD) or Raman spectroscopy to complement the SEM and AFM analyses. Verification of the sensor's performance using independent measurement techniques to ensure robustness.

Authors' response: We thank the reviewer for your insightful suggestion. We have taken XRD and XPS for 15 nm Pd₆₃Co₃₇ thin films as shown in Figure R7 and Figure R8, respectively.

Thermal evaporated C₆₀ thin film on non-epitaxial substrates like glass often leads to a mix of polycrystalline and amorphous-like regions.⁷ Thus, no peak was observed in XRD spectrum of 20 nm C₆₀ thin film in Figure R7a, confirming the absence of long-range crystalline order.³⁶ The C₆₀ film is not perfectly close-packed but exhibits partial ordering with voids and grain boundaries. It is supported by Samad *et al.* where the porosity of C₆₀ thin films was reported to be more than 50% when the substrate was kept at room temperature during the deposition.³⁷

PdCo thin films deposited on different coated substrates exhibit the strongest Bragg diffraction peaks between 41.3° and 41.6° , which can be attributed to the (111) orientation of the face-centered cubic (fcc) PdCo lattice (**Figure R7b**). These peaks are shifted to higher angles relative to the (111) peak of pure fcc Pd ($\sim 40.2^\circ$), indicating that Co atoms are substitutionally incorporated into the Pd lattice. The observed peak positions and extracted lattice constants are consistent with prior literature report by Morgan *et al.*³⁸ Analysis of the full width at half maximum (FWHM) values allowed for the estimation of crystallite size and lattice strain, as summarized in the accompanying **Table R2**. Notably, the C_{60} /PdCo sample exhibits a larger crystallite size and reduced lattice strain compared to PdCo grown directly on glass, likely due to the smoother growth surface provided by the C_{60} layer. Although slightly larger grains typically correlate with slower sensor response, the C_{60} /PdCo sample demonstrated enhanced sensitivity and faster response times (see main text **Fig. 2**), suggesting that gas diffusion through the nanoporous C_{60} interlayer plays a critical role. In contrast, TAF/PdCo showed the lowest 2θ value, indicative of lattice expansion due to tensile strain, which can be attributed to pinholes in the TAF film, as observed in AFM images (**Supplementary Fig. 9**). The improved sensing response of samples with TAF underneath appears to result from a combination of reduced grain size, efficient gas permeation through the underlayer,³⁹ and a lowered energy barrier for hydrogen sorption at the TAF/PdCo interfaces¹¹.

Figure R7. XRD spectra of **a** 20 nm C_{60} thin film and **b** 15 nm $Pd_{63}Co_{37}$ thin films on different coated glass substrates.

Table R2. Crystal structure and other physical properties of 15 nm $Pd_{63}Co_{37}$ thin films on different coated glass substrates extracted from XRD.

Sample	Peak position	FWHM	Crystallite size	Lattice Strain	d-spacing	Lattice constant
	2θ (deg)	β (deg)	D (nm)	ϵ (%)	d (Å)	a (Å)
PdCo	41.559 ± 0.002	0.93 ± 0.01	9.14 ± 0.06	1.07 ± 0.01	2.17 ± 0.01	3.76 ± 0.03
C_{60}/PdCo	41.571 ± 0.003	0.90 ± 0.01	9.55 ± 0.07	1.02 ± 0.01	2.17 ± 0.02	3.76 ± 0.03
TAF/PdCo	41.364 ± 0.007	1.25 ± 0.02	6.80 ± 0.10	1.45 ± 0.01	2.18 ± 0.03	3.78 ± 0.06

Figure R8. X-ray photoelectron spectroscopy (XPS) analysis of PdCo thin films with and without molecular coatings. **a** Full survey spectra for 15 nm PdCo (black), PdCo/2 nm C_{60} (blue), and PdCo/2 nm TAF (red). **b–d** High-resolution spectra of Co 2p, Pd 3d, and C 1s regions respectively, highlighting the changes in chemical states and bonding environments upon C_{60} and TAF coating. **e–g** Deconvoluted C 1s spectra for bare PdCo, PdCo/ C_{60} , and PdCo/TAF, respectively.

To investigate the surface modification and gain deeper insight into chemical bonding and electronic interactions between the PdCo layer and molecular coatings, we conducted X-ray photoelectron spectroscopy (XPS) analysis of PdCo thin films with and without surface functionalization, as shown in **Figure R8**. The full survey spectra (**Figure R8a**) reveal characteristic peaks corresponding to Pd 3d, Co 2p, C 1s, O 1s, and F 1s. A comparative analysis of Co 2p (**Figure R8b**) and Pd 3d (**Figure R8c**) regions between the bare PdCo and PdCo/ C_{60} samples shows negligible differences. No peak shifts or changes in the peak-to-peak intensity ratio were observed, indicating minimal impact on the Pd and Co core levels

upon C₆₀ deposition. In the PdCo/TAF spectrum, the Co 2p and Pd 3d intensities are markedly reduced, consistent with attenuation effects arising from the overlying TAF layer, which reduces both incident X-ray penetration and photoelectron emission due to its insulating character.

The C 1s region, however, displays the most pronounced differences among the samples (**Figure R8d**). Deconvoluted C 1s spectra for PdCo, PdCo/C₆₀, and PdCo/TAF are shown in **Figure R8e-g**, respectively. All samples were exposed to ambient laboratory conditions for several months; thus, the C 1s signal in bare PdCo shows typical adventitious carbon contamination, including peaks corresponding to C–C (~284.6 eV), C–O (~286.3 eV), and C=O (~288.2 eV), likely due to atmospheric CO₂ or CO adsorption.²⁹

Upon deposition of a 2 nm C₆₀ layer, the C 1s intensity increases significantly, and the dominant peak at ~284.6 eV remains, consistent with the sp²-hybridized carbon bonds of C₆₀ molecules.⁴⁰ Additional weak features in the 288–292 eV range (**Figure R8f**) suggest the emergence of interfacial bonding environments such as Pd–C or Co–C.⁴⁰ Combining these insights with the results presented in **Supplementary Fig. 13** (see page R5), we note that although bonding occurs at the C₆₀/PdCo interface, the catalytic active Pd sites for H₂ adsorption are not fully passivated. As a result, the sensor with the C₆₀ top layer exhibits comparable sensing performance to the uncoated counterpart. This indicates that C₆₀ does not significantly alter the activation energy for H₂ adsorption. Instead, the fast diffusion of H₂ molecules through the porous C₆₀ matrix appears to be the dominant factor contributing to the preserved sensing kinetics.

In the case of TAF-functionalized PdCo (**Figure R8g**), the PdCo/TAF spectrum exhibits a distinct F 1s peak at ~688 eV and its C 1s spectrum displays complex features spanning 284–293 eV. In addition to the typical C–C and C=O peaks, strong peaks at ~290.7 eV and ~293.3 eV correspond to CF₂ and CF₃ moieties, respectively, confirming the formation Pd–CF_x bonds.⁴¹ As a result, TAF coating modifies the Pd surface chemistry, lowering the activation energy for H₂ adsorption and thereby enhancing sensor responsiveness.¹¹

Change made to the manuscript:

- XRD and XPS analysis have been added to **Supplementary Note 5**.
- XRD and XPS measurements have been added to the Methods section of the manuscript (**page 28, from line 11**).

1.9. The addition of Teflon AF and PMMA layers significantly influences sensor performance. Discuss how the thickness and uniformity of these layers impact key metrics and whether these coatings introduce any limitations.

Authors' response: We thank the reviewer for the valuable comments. In this work, we mainly focus on optimizing the sensing element PdCo and layered structure, thus we have not performed the thickness dependent of the polymer layers on the final sensing performance, and the thickness of Teflon AF (TAF) and PMMA are chosen purely based on the previous works.^{1,5,19} Nevertheless, such choices are based on the following reasons.

First, the evaporated TAF thin film is also porous, like C₆₀. Additionally, TAF can significantly improve the kinetics of an H₂ sensor by modifying surface chemical and electronic states at the TAF/Pd interface, thus it is purely interfacial effect, not the bulk effect.¹¹ As a result, a thick coating layer is not required as long as it can cover the nanostructures, and a 30-nm TAF layer is sufficient.^{1,11,19} In contrast, since the PMMA coating is used to inhibit the permeation of disparate gases such as water and oxygen, its thickness is crucial. Here, we chose to coat the device with a 50-nm thick PMMA. The diffusion speeds of H₂ gas in PMMA and TAF were found to be ~1.5 μm/s and ~23 μm/s, respectively.³⁹ Therefore, in principle, a layer of <0.1 μm of PMMA coating and <1 μm of PTFE coating should not have a noticeable effect on the response time at high H₂ concentration regime. To illustrate this point, we present the sensing performance of CHA₄₅₀ sensors with and without PMMA coating in **Figure R9**. At low H₂ concentration where the external diffusion and dissociation become the rate-determining steps, coatings of any thicknesses might affect the reaction kinetics.⁴² In this context, there is a trade-off between response time and selectivity when using polymers as a coating layer.

Figure R9. Response and release time of 20 nm C₆₀/3 nm TAF/5 nm PdCo/TAF CHA₄₅₀ with and without PMMA coating.

Change made to the manuscript:

- **Figure R9** has been added to the Supplementary Information as **Supplementary Fig. 30**.
- Manuscript (page 22, from line 14) has been revised accordingly.
- Reference has been added to the manuscript:

50 Östergren, I. *et al.* Highly Permeable Fluorinated Polymer Nanocomposites for Plasmonic Hydrogen Sensing. *ACS Applied Materials & Interfaces* **13**, 21724-21732 (2021).
<https://doi.org:10.1021/acsami.1c01968>

1.10. Provide a clear section on potential future work, focusing on enhancing selectivity further and addressing interference from other gases or environmental factors.

Authors' Response: We appreciate the reviewer's suggestion.

In this study, we demonstrated the performance of our PdCo nano resistor networks under both vacuum and flow conditions with selected interferents. As future work, we aim to enhance selectivity by optimizing surface coatings that serve both as protective barriers and as selective filters for interfering species. To mitigate interference from gases such as O₂, we plan to investigate selective polymer coatings. For instance, polyvinyl alcohol (PVOH) has shown promise in suppressing O₂ interference as demonstrated by Nugroho *et al.*²³ However, due to its water solubility, PVOH is not effective under humid conditions. This highlights the need for further studies on coating thickness and tandem layer configurations to balance selectivity and environmental robustness.

Change made to the manuscript:

- This paragraph has been added to the manuscript (page 24, from line 2).
- Reference has been added to the manuscript:

59 Gaaz, T. S. *et al.* Properties and Applications of Polyvinyl Alcohol, Halloysite Nanotubes and Their Nanocomposites. *Molecules* **20**, 22833-22847 (2015).

Responses to Comments from Reviewer #2

First, I would like to congratulate the authors for such an impressive work. I follow closely the work from the group and it is no surprise that they achieve this. The sensor reported accomplishes an unprecedented combination of low limit of detection and high speed. The authors assigned the success to the addition of C₆₀ to PdCo nanohole arrays, a platform that the group develop and work a lot on, that provides extra surfaces yet interact minimally with the active element, allowing it to expand/contract freely during sensing. This conclusion (if true, see my comments below) will inspire other strategies aiming for similar goal. The reported methods are detailed, and the figure/discussion are clear. I would love to see this published in Nat Com, however, only if the authors can clarify the major questions/suggestions below. (Apology for my comments below that are random both in terms of importance and order they refer to in the text).

Authors' response: We thank the reviewer for your kind words. We are happy to have an opportunity to address the reviewer's critical remarks, important questions, and insightful suggestions.

2.1. One missing, critical point in this work (that a Nat Com paper should have) is the general insight that later can be extended to or inspire other fields. Here, author use C₆₀, but no fundamental design/understanding derived. So further characterization/model are required to back up the argument. Also, what other materials may be used to invoke similar effect?

Authors' response: We thank the reviewer for this insightful comment. We fully agree that a high-impact study should provide general design principles that can inspire future research beyond the specific materials system investigated. In our response to **Reviewer 1, Comment 1.3**, we elaborate on the multifunctional role of the C₆₀ layer and frame its function within a broader materials design strategy. In addition, we have characterized the X-ray diffraction (XRD) to study the crystallographic structures and X-ray photoelectron spectroscopy (XPS) analysis to study the chemical bonding and electronic interactions between the PdCo and C₆₀ previously shown in **Comment 1.8**.

To achieve similar effects, an interfacial layer should meet the following criteria:

- (i) enable rapid H₂ permeability without forming strong or irreversible bonds with H₂ at ambient conditions,
- (ii) maintain mechanical and chemical stability under typical environmental conditions, and
- (iii) be compatible with scalable, reproducible nanofabrication techniques, such as thermal evaporation.

These design principles point to a broader class of candidate materials beyond C₆₀. For example, porous carbon-based frameworks (e.g., carbon nanotubes²⁶), certain metal–organic frameworks (MOFs) with weak hydrogen binding energies (e.g. Zeolite imidazole framework (ZIF)⁴³), and thin polymer layers with high gas permeability and low surface energy (e.g. polyimide (PI)¹⁰ and PTFE or Teflon AF) could all potentially be engineered to deliver similar performance enhancements. We believe this framework can guide future efforts to tailor surface/interface layers in a wide range of chemical sensing platforms.

Change made to the manuscript:

- A paragraph has been added to the end of the discussion section of the manuscript (**page 26, line 16**).
- References have been added to the manuscript:

- 54 Verma, N., Delhez, R., van der Pers, N. M., Tichelaar, F. D. & Böttger, A. J. The role of the substrate on the mechanical and thermal stability of Pd thin films during hydrogen (de)sorption. *International Journal of Hydrogen Energy* **46**, 4137-4153 (2021). <https://doi.org/https://doi.org/10.1016/j.ijhydene.2020.10.163>
- 60 Darabpour, M. & Doroodmand, M. M. Fabrication of a Glow Discharge Plasma-Based Ionization Gas Sensor Using Multiwalled Carbon Nanotubes for Specific Detection of Hydrogen at Parts Per Billion Levels. *IEEE Sensors Journal* **15**, 2391-2398 (2015). <https://doi.org/10.1109/JSEN.2014.2369738>
- 61 Lv, R., Zhang, Q., Wang, W., Lin, Y. & Zhang, S. ZnO@ZIF-8 Core-Shell Structure Gas Sensors with Excellent Selectivity to H₂. *Sensors* **21**, 4069 (2021). <https://doi.org/10.3390/s21124069>

2.2. Related to that note, I find the argument explaining the reasons behind the enhancement to be unjustified and at times contradictory.

- (i) C60 is thought to increase the SVR. How? Surface here only deals with the material that interacts with H₂, exhibits properties change, which in turn converts into signal in the sensor reading. In fact, in the calculation authors include the surface of C60 but exclude its volume. Both S and V must be evaluated from the same system. Or am I wrong here? Another point for my argument is since authors said C60 does not interact with H₂, it is then as if it wasn't there, which then should not be included in the surface area consideration.

Authors' response: We thank the reviewer for the opportunity to clarify our calculation.

For PdCo deposited directly on glass substrate, the bottom surface area of PdCo is passive to the sensing process. While on C₆₀-coated, H₂ can diffuse through the voids between C₆₀ molecules, especially if the film is amorphous or contains grain boundaries (also see Comment 1.8).⁸ Thus, the bottom of the PdCo layer now is not completely passive to hydrogenation and should be considered in the surface area calculation.

The reviewer is correct that our sensing element is PdCo, thus we only need the surface and volume of PdCo. However, since it is deposited on C₆₀, the shape and thickness of PdCo are affected by the shape of its C₆₀ template and the size of PS beads as illustrated on Supplementary Fig. 15. The surface area and volume of C₆₀ layer used in the SVR calculation are just for extracting these parameters of the PdCo. For instance, top surface of C₆₀ is equal to the bottom surface of PdCo, and the volume of PdCo layer $V_{\text{PdCo}} = V_{\text{total (both bilayers)}} - V_{\text{C60}}$ from simulation point of view.

As correctly noted by the reviewer, the hydrogen absorption of C₆₀ is negligible under our testing conditions. However, we hypothesize that the C₆₀ interlayer facilitates hydrogen diffusion at the PdCo/C₆₀ interface, allowing hydrogen to access the backside of the PdCo layer, enhancing the effective diffusion and reaction area of the sensing layer.

- (ii) Even if the assumption by authors is correct, how come C60 is connected to SVR even with the thin film configuration (Fig 2). Here the only access for H₂ to PdCo is only through the top (the PdCo surface), since H₂ access through the bottom of PdCo is only through the side walls of C60, which is much smaller than the sample lateral area.

Authors' response: We thank the reviewer for this important comment, as it rightly points out that hydrogen diffusion speed in solid-state materials such as C₆₀ solid is finite and should not be overlooked. In addition, the thickness of the C₆₀ might be critical in the thin film-based sensing

regime. In response, we have conducted a comparative analysis using several additional thin-film sensor configurations and found that the hydrogen diffusion speed in C₆₀ is indeed not infinite. We also note that the response time and sensitivity of the sensor depend not only on the SVR but also on other factors such as the clamping effect and C₆₀ morphology (see Comment 1.3). We are actively pursuing direct measurements of hydrogen diffusion through C₆₀/Pd bilayer thin films locally at different positions on the films using optical techniques.¹⁴ Although these measurements are beyond the scope of the current study, the results will be reported in future work.

To improve the clarity and consistency of our manuscript, we have updated **Figure 2** in the main text. It now directly compares the sensing performance between glass/PdCo CHA and glass/C₆₀/PdCo CHA structures, rather than the thin-film structures where the hydrogen diffusion time in amorphous C₆₀ is negligible over a diffusion path length of a few hundred nanometers. The results reaffirm our findings, showing significantly enhanced sensor performance with the C₆₀ interlayer.

- (iii) C₆₀ is also thought to increase sensitivity (i.e. through hydrogen loading) due to the minimum clamping between it and the PdCo. However, author also argues that C₆₀ increases the PdCo wettability. These two are contrary to each other. So which one is true? I think authors should study in detail this interface and see if there is any chemical bonding formation.

Authors' response: Thank you for this insightful comment. Please refer to Comment 1.3 for further discussion on the role of the C₆₀ layer, including the clamping effect and its impact on sensing performance.

We would like to clarify our original statement: PdCo exhibits improved wettability on C₆₀ compared to glass, but this does not contradict the observation of a weaker clamping effect. This distinction addresses your concern regarding the seemingly conflicting interpretations of wettability and mechanical decoupling.

Glass has inherently low surface energy and is chemically inert, leading to poor adhesion and discontinuous film formation for PdCo. In our previous work, Pham *et al.* observed that PdCo films thinner than 10 nm deposited on glass tend to form isolated islands, resulting in high electrical resistance and even negative hydrogen sensitivity due to nanoparticle coalescence during hydrogenation.⁵ Similar behavior has been reported in the literature.^{44,45}

When deposited on C₆₀, Pd and Co atoms exhibit slightly improved wettability, which we attribute to weak charge-transfer interactions particularly under low surface coverage or elevated temperature.⁴⁶ However, these interactions remain weak, allowing for a loosely bonded interface with abundant free Pd at the surface, beneficial for hydrogen catalysis. Film growth still follows the Volmer–Weber (island) mode, resulting in a discontinuous morphology. This interpretation is supported by prior studies. For instance, Kaushik *et al.* report a lower percolation threshold for Co on C₆₀ (2.3–4.0 nm) than on glass, indicating better film formation without strong bonding.⁴⁷ Similarly, Wang *et al.* demonstrated that Pd (3 nm)/C₆₀ (3 nm) multilayers form small, high-resistance Pd islands.⁴⁸ These findings suggest that while PdCo spreads more readily on C₆₀ than on glass, the interfacial interaction remains weak and mechanically decoupled.

Regarding chemical bonding, direct spectroscopic evidence is subtle due to the weak nature of Pd–C₆₀ interactions. For detailed discussion on these interactions and their detection via techniques such as X-ray photoelectron spectroscopy (XPS), we refer to previous works by Weckesser *et al.*⁴⁶ and Anděra *et al.*⁴⁰ Furthermore, we provide evidence of chemical bonding between C₆₀ and Pd by XPS analysis of the PdCo/C₆₀ bilayer, supporting the interfacial interaction hypothesis (see Comment 1.8).

In summary, PdCo films display better morphology on C₆₀ than on glass due to improved wettability, but the chemical interactions remain weak. Since C₆₀ film itself is composed of van der Waals-bound molecules, the PdCo and C₆₀ layers remain largely decoupled, ensuring low clamping forces during hydrogenation and dehydrogenation, thereby, preserving and enhancing sensor performance.

- (iv) Further contrasting argument related to point (iii) is that such enhancement effect is also present when there is FTA in between C₆₀ and PdCo. So what is the role of C₆₀ here then? Does FTA also have similar effect as C₆₀? If so, why then it is thickness-dependent? And the thickness dependency also shows rather contradicting results. If it really due to clamping being less, the thicker the FTA, the less the clamping effect is. However, for 50 nm, the thickest FTA studied, the improvement is actually the lowest. All in all there are major investigation and analysis required to reach any conclusion for the study.

Authors' response: We thank the reviewer for this insightful comment. To clarify, we have only conducted thickness dependence of C₆₀ and have not done so for TAF. As shown in **Figure 3**, the response time and sensitivity of the CHAs appear independent of C₆₀ thickness. This may be because the CHA morphology already introduces abundant nanoscale pathways and adsorption sites for H₂, effectively masking any subtle influence from varying C₆₀ thickness within the tested range.

TAF, however, cannot fully substitute for C₆₀. Due to poor adhesion to the glass substrate, TAF alone is completely removed during the polystyrene bead lift-off process, and PdCo might penetrate into a soft TAF layer resulting in a discontinuous network. The C₆₀ layer serves as a necessary morphological template, defining the geometry of the PdCo nanostructure. When only a few-nm TAF is deposited on top of the C₆₀, it provides three synergistic advantages: (i) preserving of structure as the underlying C₆₀ geometry is retained, (ii) PdCo is surrounded by TAF, which is known to promote hydrogen adsorption kinetic,¹¹ and (iii) the underlayer of TAF induces lattice strain and reduces PdCo crystallite size as shown in XRD (**Figure R7**, page R17), which can enhance hydrogen sorption kinetics.

Change made to the manuscript:

- **Figure 2** in the main text has been updated.
- The role of C₆₀ has been elaborated in the main text (page 10, from line 1; page 11, from line 5; page 13, from line 10; page 15 from line 1) and the role of TAF has been also detailed in the manuscript (page 21, from line 16).

2.3. Results and Discussion headers are missing.

Authors' response: We thank the reviewer for pointing it out. We have added the headers and updated the manuscript following the journal's format guidelines.

2.4. Please discuss recent strategies in improving sensitivity (to ppb level) in H₂ sensors (electric, optic, ect). This will benefit the readers immensely.

Authors' response: We thank the reviewer for your constructive suggestion. We have added **Table R1** (page R8) with state-of-the-art ultra-low LOD hydrogen sensors across all platforms (e.g. electrical, optical, etc.), including the devices' structure, transducing methods and their corresponding sensing metrics for a broader comparative context. Additionally, the Introduction in the manuscript has been revised accordingly to discuss this point.

Change made to the manuscript:

- **Table R1** has been added to the Supplementary Information as **Supplementary Table 1**.
- The following paragraph has been added to the introduction of the manuscript (**Page 3, Line 8**).

“Recent advancements across both electrical and optical platforms in achieving ultra-fast response time and ppb-level sensitivity in hydrogen sensors have centered on nanostructured Pd hydride systems due to their high selectivity, excellent sensitivity, fast response times, and mechanical and chemical robustness (see Supplementary Table 1).^{4,5,11,19,31,35,49} Pd-based alloys (e.g., PdCo, PdAu) dominate high-impact designs. For instance, recent works on optical sensors by Nugroho *et al.* using scattering spectrum from PdAu nanoparticle@polymer nanocomposite materials¹, and Luong *et al.* using transmission intensity through PdCo nanocap arrays⁵ explicitly demonstrate sub-second response with LODs around single-digit ppm. The LOD could be further improved to a few hundreds of ppb by making use of the particle swarm optimization algorithm to optimize the sensor array configurations,⁵⁰ neural-network-based data treatment to correct the baseline drift while testing in extreme humidified conditions,⁵¹ or by simply stacking the sensors vertically to improve the signal-to-noise ratio (SNR).¹⁹

In electrical platforms, this transducing method shows great promise for achieving ultra-sensitive LOD of a few ppb by virtue of the low noise signal readouts. Pham *et al.* proposed a PdCo composite hole array (CHA) nano network with a measured LOD of 180 ppb, however the response time fell short of the 1-s response time benchmark required for automotive applications.⁵ Pd-doped oxides (e.g., Pd@Fe₂O₃ nanotubes) combine Pd's catalytic activity with oxide heterojunctions, achieving 50 ppb LOD through synergistic surface reactions and p–n junction modulation.¹⁸ Non-Pd materials, such as rGO/ZnO-SnO₂ composites, also reach 50 ppb sensitivity via Pd doping and heterostructure engineering,²² though they lag behind Pd hydrides in speed and ambient stability. These strategies highlight the critical role of nanostructuring, alloying, and hybrid material design tailoring for each transducing method and sensing mechanism in pushing sensitivity limits, with Pd-based systems remaining the gold standard for safety-critical applications.”

- References have been added to the main text:

- 14 Tomeček, D. *et al.* Neural network enabled nanoplasmonic hydrogen sensors with 100 ppm limit of detection in humid air. *Nature Communications* **15**, 1208 (2024). <https://doi.org/10.1038/s41467-024-45484-9>
- 15 Mo, T. *et al.* High Response and ppb-Level Detection toward Hydrogen Sensing by Palladium-Doped α -Fe₂O₃ Nanotubes. *ACS Sensors* **9**, 5976-5984 (2024). <https://doi.org/10.1021/acssensors.4c01829>
- 16 Zhang, X. *et al.* Ultralow detection limit and ultrafast response/recovery of the H₂ gas sensor based on Pd-doped rGO/ZnO-SnO₂ from hydrothermal synthesis. *Microsystems & Nanoengineering* **8** (2022). <https://doi.org/10.1038/s41378-022-00398-8>

2.5. In the real setting, is there a need to measure H₂ down to ppt level?

Authors' response: We thank the reviewer for raising this important point. While the U.S. Department of Energy (DOE) specifies a detection requirement of 10 ppb for hydrogen in environmental monitoring, there are compelling reasons to pursue detection limits down to the low ppb or even ppt range.

First, pushing the detection limit serves as a benchmark for optimizing sensor performance under noise-limited conditions. Achieving reliable detection at these ultra-low concentrations requires a low-noise transduction platform, high signal-to-noise ratios, and advanced signal processing techniques to detect early signal variations.^{51,52}

Second, sensors capable of ppt-level hydrogen detection are valuable for specialized applications such as deep-space missions, where trace hydrogen detection is critical in planetary science and astrobiology.²⁴

Finally, there is growing interest in detecting hydrogen isotopes such as deuterium and tritium, particularly in the context of nuclear power and waste management, where their concentrations can be significantly lower than that of molecular hydrogen in air.⁵³ Our sensor platform holds promise for further development to detect these isotopes in such challenging environments.

2.6. It is mentioned that GLACD method enhances quality, morphology, and SVR. How is it so?

Authors' response: We thank the reviewer for pointing it out. We have removed that claim in the abstract and revised the introduction in the manuscript to clarify/highlight advantages of employing glancing angle co-deposition in our sensor design as follows:

GLACD enables the formation of a mixture of two metals under far-from-equilibrium conditions, resulting in films with high defect densities and internal stress, these features create abundant diffusion paths and active sites that are advantageous for sensing applications. The porous, columnar morphology typical of GLACD films further improves gas accessibility.⁵⁴ Despite being structurally less ordered, such films offer tunable properties not achievable through equilibrium processing.⁵⁵

Change made to the manuscript:

- The manuscript's abstract and introduction (page 5, line 2) have been revised accordingly.
 - References have been added to the main text:
- 25 Ai, B. & Zhao, Y. Glancing angle deposition meets colloidal lithography: a new evolution in the design of nanostructures. **8**, 1-26 (2019). <https://doi.org/doi:10.1515/nanoph-2018-0105>

2.7. Related to above, it is said the method will get the sample to “their full potential.” This is highly speculative and unjustified. Also, what are their full potential? Is there any fundamental ceiling to this?

Authors’ response: We thank the reviewer for this insightful comment. We agree that the original phrase “full potential” was overly strong and could be interpreted as speculative or unjustified. Our intention was to convey that we have undertaken a systematic optimization of key parameters such as film thickness, material concentration, and surface modification to enhance the performance of our sensing platform. In response to the reviewer’s feedback, we have revised both the abstract and main text to remove the phrase “full potential” and to avoid any implication of a fundamental performance limit. Instead, we now highlight our approach to performance optimization without making speculative claims about ultimate material capabilities.

Change made to the manuscript:

- The term “their full potential” has been removed.
- The abstract and introduction of the manuscript have been revised accordingly.

2.8. Since the LOD is the result of extrapolation, I don’t think it is appropriate to use it in the title and abstract, as it may trick readers. Only highlight such a number in these places if it is demonstrated, or explicitly write that it is an “extrapolated LOD”

Authors’ response: We thank the reviewer for an on-point comment. We have modified the title to claim ppb LOD, which was a measured LOD rather than the extrapolated LOD. Please refer to **Comment 2.9** below for the updated LOD data.

Change made to the manuscript:

- The title has been modified from “parts-per-trillion” to “parts-per-billion”.

2.9. Regarding the extrapolation, what is the basis for it? I think it is never mentioned. Or the authors just extend the line following the data?

Authors’ response: The reviewer is correct. This is the way that many papers have presented to extract the ultimate LOD due to the limitation of the test equipment. Here, we have taken additional LOD experiments for 3 individual devices (**Figure R10**) to verify the ultimate measured LOD of 40 ppb without any extrapolation and modified the abstract at title accordingly.

Figure R10. LOD measurements of 20 nm $C_{60}/3\text{nm TAF}/5\text{ nm PdCo}/30\text{ nm TAF CHA}_{450}$ sensor. **a, b** Sensors' response at low H_2 pressure regime. **c** Measured sensitivities across 6 orders of H_2 pressure. **d** Averaged $\Delta R/R$ plot as a function of P_{H_2} with the error bars indicate the standard deviation from 3 devices. The vertical blue line denotes the defined LOD of $3\sigma = 0.012\%$ at 8.4 Hz sampling frequency (**Supplementary Note 8**). All measurements were performed in vacuum mode at room temperature. Note that **d** is identical with Figure 6d in the manuscript.

Change made to the manuscript:

- **Figure R10** has been added to Supplementary Information as **Supplementary Fig. 27**.
- In the main text, **Figure 6d** has been replaced with **Figure R10d**.
- The title and abstract have been revised accordingly.

2.10. Again regarding extrapolation, I don't think it is justified to claim a value from an extrapolation 3 orders of magnitude higher/lower than the last experimental data point (here from 100 ppb to 0.1 ppb). Many things/trends can happen within that expansive range.

Authors' response: We thank the reviewer for raising this important point. We agree with the reviewer that the slopes of the pressure-composition curve are not the same across the wide range of H_2 pressure depending on which step is the rate-determining step. There is a need to develop a theoretical model for H_2 diffusion in Pd nanoparticles to justify the extrapolation at extremely low H_2 concentration.

In the revised manuscript, we removed the extrapolated curve and only focused on highlighting the measured LOD of 40 ppb with the signal-to-noise ratio $\text{SNR} \approx 10$ with a potential to detect a few ppb.

Change made to the manuscript:

- The title and abstract have been revised accordingly.

2.11. Another main concern is the use of idealized environments throughout the study. Looking at the state of the field now, where it has progressed immensely, I would highly encourage that we start using “real environment” more, especially for a publication in Nat Com. Thus, I am curious to see how the sensors respond in (synthetic) air. This is important in general, but even more for the authors since they claim that their sensors fulfil the strictest requirement for environmental monitoring (which ironically comprises air as background). Achieving what the authors did is impressive, but this does not reflect the real condition at all. In air, there will be many complex reactions being involved. How does the proposed concept perform in air? If not, how far is it from the ideal situation? As such, similar to my comment above, whenever ideal conditions are used, please also have an explicit statement regarding it when mentioning the performance (especially in title and abstract).

Authors’ response: We appreciate the reviewer’s thoughtful comment and fully agree that testing under realistic conditions (i.e. in air) is essential for evaluating the practical utility of hydrogen sensors. The presence of oxygen in air can indeed alter surface chemistry and sensor performance by blocking Pd active sites and promoting side reactions such as water formation through the oxygen reduction reaction (ORR).⁵⁶ This remains an area of active and ongoing investigation. Here, we incorporated recent advances in surface passivation strategy proposed by Nugroho *et al.*, which employs a tandem coating consisting of PMMA and only 5-nm thin PVOH (polyvinyl alcohol) layers, where the PVOH coating effectively mitigates the O₂ permeability while maintaining high H₂ diffusivity.²³ Our preliminary results of 15 nm Pd₆₇Co₃₃ thin films are presented in **Figure R11**.

Comparing the signal change to 2% H₂ in synthetic air versus in N₂ as gas carrier (**Figure R11a**), the baseline drift may be attributed the faster hydrogen desorption in the presence of O₂ in air and eventually forming water via ORR.⁵⁶ While a 50 nm PMMA layer offers limited protection (**Figure R11b**), PVOH coatings effectively suppress the adverse effects of oxygen shown in the absence of such baseline drifts when the sensors coated with PVOH (**Figure R11c-d**). As demonstrated in Gaaz *et al.*, polyvinyl alcohol exhibits superior oxygen barrier properties among polymers due to its dense hydrogen bonding network and partial crystallinity, making it an effective shielding layer to suppress O₂ interference on active sensing surfaces under dry conditions.⁵⁷ However, even with just a few nanometers of coverage, the PVOH layer significantly reduces hydrogen sorption kinetics. While the PMMA-coated sensor exhibits a response time t_{90} of 1.9 ± 0.1 s (extracted from the flow in N₂), the addition of 8 nm and 4 nm PVOH coatings increases t_{90} to 31.0 ± 0.5 s and 22.0 ± 0.2 s, respectively, highlighting a strong diffusion barrier effect for H₂. Additionally, due to its water solubility, PVOH is not effective under humid conditions. This highlights the need for further studies on coating thickness and tandem layer configurations to balance selectivity, detection speed and environmental robustness, which is beyond the scope of this study.

Figure R11. Sensing response of 15 nm PdCo thin film sensors with different polymer coatings to 2% H₂ in N₂ and synthetic air as gas carrier. All measurements were performed in flow mode with a constant flow rate of 400 sccm at room temperature.

Change made to the manuscript:

- The testing conditions have been listed explicitly in the last paragraph of the manuscript introduction (page 5, lines 13 and 22).
- The following discussion has been added to the manuscript (page 24, line 3):
 “While this study focuses on idealized conditions, we acknowledge that evaluating sensor performance in synthetic air is essential for real-world applications. Some preliminary results with polyvinyl alcohol (PVOH)-coated sensors show that oxygen interference can be mitigated; however, the coating introduces trade-offs in hydrogen diffusion and limits applicability under humid conditions.^{23,57} Further optimization of coating thickness and architecture to balance selectivity, response time, and environmental robustness remains an important future direction beyond the scope of this work.”
- References have been added to the main text:
 59 Gaaz, T. S. *et al.* Properties and Applications of Polyvinyl Alcohol, Halloysite Nanotubes and Their Nanocomposites. *Molecules* **20**, 22833-22847 (2015).

2.12. Related to above, in the contamination tests, I encourage authors to instead expose the sensors to constant dose of contaminations, instead of having them in cycle together with the hydrogen. This, again, is closer to the real conditions. I am curious to see the response to humidity using this scenario, as based on my experience, PMMA would absorb water and eventually deteriorates the sensor's performance.

Authors' response: We appreciate the reviewer's insightful suggestion to cycle the sensor through varying levels of humidity while monitoring the response, as it would indeed better reflect real-world conditions. However, due to the design of our current experimental apparatus, where humidity is introduced via a water bath humidifier, we are only able to maintain fixed humidity levels during the tests. Unfortunately, this setup does not allow cycling across multiple RH levels within a single measurement sequence at the moment.

We acknowledge this as a limitation of our current work. Nonetheless, we believe our measurements at two representative conditions provide meaningful insight into the sensor's performance. We consider dynamic RH cycling an important direction for future apparatus improvements.

Responses to Comments from Reviewer #3

The authors described how to fabricate unique hydrogen sensing element, a PdCo composite hole array on a fullerene-decorated glass substrate, and evaluated the sensing times and detection limits for H₂ concentrations were evaluated in relation to the structure of the sensing element. By fine-tuning the structure of the PdCo composite hole array, they achieved a reaction time of less than 1 second and a detection limit at ppt level. It is noteworthy that a high-performance sensor element can be constructed using a simple fabrication method, and the structure of this element may also be applicable to gas sensors other than hydrogen. However, I believe that further discussion is needed on the relationship between the structure and performance of sensing elements before publication, the authors need to fully and in detail address the following points:

Authors' response: We thank the reviewer for your kind words. We are happy to have an opportunity to address the reviewer's critical remarks, important questions, and insightful suggestions.

3.1. The authors need to explain the crystal structures of the PdCo and C₆₀ layers, as this information is crucial for understanding the hydrogen absorption properties of the PdCo layer and the H₂ gas behavior through the C₆₀ layer.

Authors' response: We thank the reviewer for your insightful suggestion. We have taken XRD for 15 nm Pd₆₃Co₃₇ thin films deposited on different coated glass substrates as shown in **Figure R7**. XRD was taken using X-ray diffractometer (Rigaku, Cu K α : $\lambda = 1.54056 \text{ \AA}$, 40 kV, and 40 mA). First, thermal evaporated C₆₀ thin film on non-epitaxial substrates like glass often leads to a mix of polycrystalline and amorphous-like regions.⁷ Thus, no peak was observed in XRD spectrum of 20 nm C₆₀ thin film in **Figure R7a**, confirming the absence of long-range crystalline order.³⁶ The C₆₀ film is not perfectly close-packed but exhibits partial ordering with voids and grain boundaries. It is supported by Samad *et al.* where the porosity of C₆₀ thin films was reported to be more than 50% when the substrate was kept at room temperature during the deposition.³⁷

Second, PdCo thin films deposited on different coated substrates exhibit the strongest Bragg diffraction peaks between 41.3° and 41.6°, which can be attributed to the (111) orientation of the face-centered cubic (fcc) PdCo lattice. These peaks are shifted to higher angles relative to the (111) peak of pure fcc Pd (~40.2°), indicating that Co atoms are substitutionally incorporated into the Pd lattice. The observed peak positions and extracted lattice constants are consistent with prior literature report.³⁸ Analysis of the full width at half maximum (FWHM) values allowed for the estimation of crystallite size and lattice strain, as summarized in the accompanying **Table R2**. Notably, the C₆₀/PdCo sample exhibits a larger crystallite size and reduced lattice strain compared to PdCo grown directly on glass, likely due to the smoother growth surface provided by the C₆₀ layer. This finding supports our claim of the improvement in wettability between Pd and the C₆₀-coated substrate. Although larger grains typically correlate with slower sensor response, the C₆₀/PdCo sample demonstrated enhanced sensitivity and faster response times (**Fig. 2** main text), suggesting that gas diffusion through the C₆₀ interlayer plays a critical role. In contrast, TAF/PdCo showed the lowest 2 θ value, indicative of lattice expansion due to tensile strain, which can be attributed to pinholes in the TAF film, as observed in AFM imaging (**Supplementary Fig. 9**). The improved sensing response of samples with TAF underneath appears to result from a combination of

reduced grain size, efficient gas permeation through the underlayer, and a lowered energy barrier for hydrogen sorption at the TAF/PdCo interfaces¹¹.

Figure R7. XRD spectra of **a** 20 nm C₆₀ thin film and **b** 15 nm Pd₆₃Co₃₇ thin films on different coated glass substrates.

Table R2. Crystal structure and other physical properties of 15 nm Pd₆₃Co₃₇ thin films on different coated glass substrates extracted from XRD.

Sample	Peak position	FWHM	Crystallite size	Lattice Strain	d-spacing	Lattice constant
	2θ (deg)	β (deg)	D (nm)	ε (%)	d (Å)	a (Å)
PdCo	41.559 ± 0.002	0.93 ± 0.01	9.14 ± 0.06	1.07 ± 0.01	2.17 ± 0.01	3.76 ± 0.03
C ₆₀ /PdCo	41.571 ± 0.003	0.90 ± 0.01	9.55 ± 0.07	1.02 ± 0.01	2.17 ± 0.02	3.76 ± 0.03
TAF/PdCo	41.364 ± 0.007	1.25 ± 0.02	6.80 ± 0.10	1.45 ± 0.01	2.18 ± 0.03	3.78 ± 0.06

Change made to the manuscript:

- XRD analysis has been added to Supplementary Note 5.
- XRD and XPS measurements have been added to the Methods section of the manuscript (page 28, from line 11).

3.2. The effect of the C60 layer is unclear. There seem to be a contradiction between the following two considerations. In fig. 2, the authors pointed out that the C60 layer mitigated the clamping effect on the PdCo layer, maximizing the hydrogen uptake and the sensitivity. Meanwhile, in fig. 6, the TAF layer inserted between the PdCo and C60 layers was useful to improving sensitivity. If the TAF layer also works to mitigate the clamping effect, the effect of C60 is thought to be only due to its weak adhesion to the substrate.

Authors' response: Thank you for this insightful comment. We note that **Reviewers 1** and **2** raised similar concerns; for a detailed discussion of the role of the C₆₀ layer and its influence on sensing performance, please refer to our response to Comment 1.3. The clamping effect is addressed in Comment 2.2 (iii) on page R24, and the role of the TAF layer is discussed in Comment 2.2 (iv) on page R25.

3.3. What about temperature stability in resistance measurement? Since resistance in metals depends on temperature, it is possible that the baseline of resistance $R(0)$ can shift depending on temperature. While the baseline of $\Delta R/R$ shown in figs. S18 and S21 of the supplementally materials shows an approximately straight line, the resistance was not shown. I kindly ask the authors to comment on the stability of the temperature of the system during measurements and on the temperature dependence of the resistance for the fabricated sample.

Authors' response: We thank the reviewer for this valuable comment. To address the temperature stability of the system and the temperature dependence of resistance, we conducted additional measurements at varying temperatures from room temperature ($\sim 23^\circ\text{C}$) up to 115°C . The temperature was continuously monitored using a thermocouple affixed to the substrate holder inside the gas cell. As shown in Figure R5a, the system exhibited excellent thermal stability, with fluctuations remaining below 0.3°C throughout the measurement period.

In Figure R5b, we present the resistance response under hydrogen exposure across the tested temperature range. The baseline resistance R_0 values at each temperature are shown in Figure R5c. At lower temperatures, R_0 remains highly stable, following an approximately linear increase from $1260\ \Omega$ at room temperature to $\sim 1320\ \Omega$ at 100°C , consistent with typical metallic behavior where increased electron–phonon scattering raises resistance. We also note that the current used for resistance measurement is limited to $0.1\ \text{mA}$, minimizing Joule heating effects. However, at 115°C , R_0 slightly decreases to $\sim 1290\ \Omega$, which we attribute to possible thermally induced morphological changes in the nanoparticle network such as expansion or coalescence that may alter percolation paths.

As illustrated in Figure R5d, the sensor's sensitivity declines at elevated temperatures, consistent with suppressed β -phase hydride formation and enhanced desorption kinetics at higher temperatures.⁴

Overall, our findings suggest that room temperature provides the optimal balance between baseline stability and sensing performance for our PdCo nano network structure. For field applications, we propose integrating a local temperature control element such as a thermoelectric cooler (TEC) into the sensor housing to ensure consistent operation under ambient fluctuations.

Figure R5. **a** Measured temperatures at the sensor position during the tests. **b** Sensing responses of 20 nm C₆₀/5 nm PdCo CHA₄₅₀ to stepwise decreasing H₂ concentrations with N₂ as the gas carrier and **c** extracted baseline R₀ and **d** extracted sensitivity. The error bars in **d** indicate the standard deviation from 3 pulses.

3.4. Why were response times measured in vacuum mode? For the same hydrogen concentration, is the response time the same in vacuum mode and flow mode? We believe that the response time in flow mode will be slower than that in vacuum mode, because the H₂ diffusion time in the carrier gas (N₂) may be the rate-limiting step for detection at low H₂ concentration in mixture gas. Furthermore, when applying samples to hydrogen sensors in atmospheric environments, we generally believe that the response time in flow mode reflects reality.

Authors' response: We thank the reviewer for this important comment. We agree that flow mode better reflects real-world deployment conditions for hydrogen sensors. However, we also believe vacuum mode measurements serve a distinct and complementary purpose, particularly in evaluating and optimizing the intrinsic response speed of the sensing material itself.

As shown in the added **Figure R12**, response time in flow mode is significantly affected by external parameters such as carrier gas flow rate. For example, reducing the total flow from 400 sccm to 32 sccm

results in a substantial increase in response time, illustrating that external diffusion becomes the rate-limiting step.⁴² In contrast, vacuum mode eliminates these transport limitations as the gas pressure reaches equilibrium in <100 ms (**Supplementary Fig. 4**) and allows us to isolate the sensing element's intrinsic kinetics.

To provide a specific comparison:

- In vacuum mode (1 mbar H₂, ~0.1%), the CHA₃₀₀ sensor reaches 90% response in under 2 s (see main text **Figure 4g**).
- In flow mode at 400 sccm, the same concentration yields a t₉₀ of ~30 s (**Figure R12b**).

This large discrepancy highlights that the gas dissociation and internal diffusion in our CHA-based sensor are inherently fast, and that the observed delay in flow mode stems from the time required for hydrogen molecules to diffuse and adsorb onto the surface of the sensing element.

Therefore, while flow mode is essential for practical validation, vacuum mode offers valuable insights into the material-level optimization and can serve as a benchmark for assessing the limitations imposed by sensor housing and gas delivery design. This comparison also raises important considerations for future device engineering, such as enhancing gas dynamics via integrated pumps, or flow channel optimization.⁵⁸

Figure R12. **a** Sensing response of 20 nm C₆₀/5 nm PdCo/TAF CHA₃₀₀ sensor to 0.1% H₂ pulses at different total flow rates and **b** extracted response times as a function of flow rate. All measurements were performed at room temperature.

Change made to the manuscript:

- The discussion regarding vacuum mode and flow mode measurements have been added to the **Supplementary Note 3**.

3.5. In fig. 7, is the deterioration speed of the uncovered sample (20nmC60/5nmPdCo/TAF CHA450) to interference gases and humidity faster than that of the PMMA-covered sample? Since the data of the uncovered sample was not shown, the effect of the PMMA coating on interference gases and humidity was not clear in the manuscript. Please show the data of the comparative sample related to fig. 7 in the supplementally materials.

Authors' response: We thank the reviewer for this valuable comment. In this study, we directly tested the PMMA-coated samples based on our previous findings in optical sensors,¹ where the uncoated PdCo sensing element exhibited a pronounced signal degradation upon exposure to 0.2% CO and RH = 40%. In the present work using an electrical resistor network, we observed a similar decline in signal amplitudes for the uncoated device under interfering gas exposure as depicted in **Figure R13**. This deterioration is attributed to the competitive adsorption of CO and H₂O molecules on the PdCo surface, which blocks active hydrogen adsorption sites. This observation is consistent with recent findings by Kim *et al.*,²⁹ who reported that CO binds strongly to Pd surfaces due to its higher adsorption energy compared to CO₂ and H₂O. To further support this, we include the XPS C 1s spectrum of the bare PdCo thin film, showing a prominent C=O bond signal at ~288 eV (see **Figure R4**, page R12), indicating CO adsorption.

Figure R13. Interference tests of 20 nm C₆₀/3 nm TAF/5 nm PdCo/TAF CHA₄₅₀ (no PMMA coating) with different **a** gases and **b** relative humidity (RH) levels. The error bars in **b** and **d** indicate the standard deviation from 10 pulses. All measurements were performed at room temperature in flow mode at a constant flow rate of 400 sccm and with N₂ as gas carrier.

Change made to the manuscript:

- **Figure R13** has been added to the Supplementary Information as **Supplementary Fig. 29**.
- The manuscript has been revised accordingly (page 22, line 15).

Rebuttal references

- 1 Luong, H. M. *et al.* Sub-second and ppm-level optical sensing of hydrogen using templated control of nano-hydride geometry and composition. *Nature Communications* **12**, 2414-2414 (2021). <https://doi.org:10.1038/s41467-021-22697-w>
- 2 Darmadi, I., Nugroho, F. A. A., Kadkhodazadeh, S., Wagner, J. B. & Langhammer, C. Rationally Designed PdAuCu Ternary Alloy Nanoparticles for Intrinsically Deactivation-Resistant Ultrafast Plasmonic Hydrogen Sensing. *ACS Sensors* **4**, 1424-1432 (2019). <https://doi.org:10.1021/acssensors.9b00610>
- 3 Bérubé, V., Radtke, G., Dresselhaus, M. & Chen, G. Size effects on the hydrogen storage properties of nanostructured metal hydrides: A review. *International Journal of Energy Research* **31**, 637-663 (2007). <https://doi.org:10.1002/er.1284>
- 4 Darmadi, I., Nugroho, F. A. A. & Langhammer, C. High-Performance Nanostructured Palladium-Based Hydrogen Sensors—Current Limitations and Strategies for Their Mitigation. *ACS Sensors* **5**, 3306-3327 (2020). <https://doi.org:10.1021/acssensors.0c02019>
- 5 Pham, M. T. *et al.* Pd80Co20 Nanohole Arrays Coated with Poly(methyl methacrylate) for High-Speed Hydrogen Sensing with a Part-per-Billion Detection Limit. *ACS Applied Nano Materials* **4**, 3664-3674 (2021). <https://doi.org:10.1021/acsanm.1c00169>
- 6 Mortazavi, B. Structural, electronic, thermal and mechanical properties of C60-based fullerene two-dimensional networks explored by first-principles and machine learning. *Carbon* **213**, 118293 (2023). <https://doi.org:10.1016/j.carbon.2023.118293>
- 7 Faiman, D. *et al.* Structure and optical properties of C60 thin films. *Thin Solid Films* **295**, 283-286 (1997). [https://doi.org:https://doi.org/10.1016/S0040-6090\(96\)09043-8](https://doi.org:https://doi.org/10.1016/S0040-6090(96)09043-8)
- 8 Saha, D. & Deng, S. Hydrogen Adsorption on Pd- and Ru-Doped C60 Fullerene at an Ambient Temperature. *Langmuir* **27**, 6780-6786 (2011). <https://doi.org:10.1021/la200091s>
- 9 Bozhko, S. I., Walshe, K. & Shvets, I. V. Control of binding of C60 molecules to the substrate by Coulomb blockade. *Scientific Reports* **9** (2019). <https://doi.org:10.1038/s41598-019-52544-4>
- 10 Verma, N., Delhez, R., van der Pers, N. M., Tichelaar, F. D. & Böttger, A. J. The role of the substrate on the mechanical and thermal stability of Pd thin films during hydrogen (de)sorption. *International Journal of Hydrogen Energy* **46**, 4137-4153 (2021). <https://doi.org:https://doi.org/10.1016/j.ijhydene.2020.10.163>
- 11 Nugroho, F. A. A. *et al.* Metal-polymer hybrid nanomaterials for plasmonic ultrafast hydrogen detection. *Nature Materials* **18**, 489-495 (2019). <https://doi.org:10.1038/s41563-019-0325-4>
- 12 Kroto, H. W., Heath, J. R., O'Brien, S. C., Curl, R. F. & Smalley, R. E. C60: Buckminsterfullerene. *Nature* **318**, 162-163 (1985). <https://doi.org:10.1038/318162a0>
- 13 Aziz, M. T., Gill, W. A., Khosa, M. K., Jamil, S. & Janjua, M. R. S. A. Adsorption of molecular hydrogen (H₂) on a fullerene (C₆₀) surface: insights from density functional theory and molecular dynamics simulation. *RSC Advances* **14**, 36546-36556 (2024). <https://doi.org:10.1039/d4ra06171c>
- 14 Chang, P.-C., Chang, Y.-Y., Wang, W.-H., Lo, F.-Y. & Lin, W.-C. Visualizing hydrogen diffusion in magnetic film through magneto-optical Kerr effect. *Communications Chemistry* **2** (2019). <https://doi.org:10.1038/s42004-019-0189-1>
- 15 Dediu, V. A., Hueso, L. E., Bergenti, I. & Taliani, C. Spin routes in organic semiconductors. *Nature Materials* **8**, 707-716 (2009). <https://doi.org:10.1038/nmat2510>
- 16 Mandal, S. *et al.* A robust organic hydrogen sensor for distributed monitoring applications. *Nature Electronics* (2025). <https://doi.org:10.1038/s41928-025-01352-y>
- 17 Askar, P. *et al.* 1 ppm-detectable hydrogen gas sensor based on nanostructured polyaniline. *Scientific Reports* **14** (2024). <https://doi.org:10.1038/s41598-024-77083-5>
- 18 Mo, T. *et al.* High Response and ppb-Level Detection toward Hydrogen Sensing by Palladium-Doped α -Fe₂O₃ Nanotubes. *ACS Sensors* **9**, 5976-5984 (2024). <https://doi.org:10.1021/acssensors.4c01829>

- 19 Luong, H. M. *et al.* Ultra-fast and sensitive magneto-optical hydrogen sensors using a magnetic nano-cap array. *Nano Energy* **109**, 108332 (2023).
[https://doi.org:https://doi.org/10.1016/j.nanoen.2023.108332](https://doi.org/https://doi.org/10.1016/j.nanoen.2023.108332)
- 20 Wen, L. *et al.* On-chip ultrasensitive and rapid hydrogen sensing based on plasmon-induced hot electron–molecule interaction. *Light: Science & Applications* **12**, 76 (2023).
[https://doi.org:10.1038/s41377-023-01123-4](https://doi.org/10.1038/s41377-023-01123-4)
- 21 Zhang, H. *et al.* 1ppm-detectable hydrogen gas sensors by using highly sensitive P+/N+ single-crystalline silicon thermopiles. *Microsystems & Nanoengineering* **9** (2023).
[https://doi.org:10.1038/s41378-023-00506-2](https://doi.org/10.1038/s41378-023-00506-2)
- 22 Zhang, X. *et al.* Ultralow detection limit and ultrafast response/recovery of the H₂ gas sensor based on Pd-doped rGO/ZnO-SnO₂ from hydrothermal synthesis. *Microsystems & Nanoengineering* **8** (2022). [https://doi.org:10.1038/s41378-022-00398-8](https://doi.org/10.1038/s41378-022-00398-8)
- 23 Nugroho, F. A. A. *et al.* Inverse designed plasmonic metasurface with parts per billion optical hydrogen detection. *Nature Communications* **13** (2022). [https://doi.org:10.1038/s41467-022-33466-8](https://doi.org/10.1038/s41467-022-33466-8)
- 24 Tian, J. *et al.* A Ppb-level hydrogen sensor based on activated Pd nanoparticles loaded on oxidized nickel foam. *Sensors and Actuators B: Chemical* **329**, 129194 (2021).
[https://doi.org:https://doi.org/10.1016/j.snb.2020.129194](https://doi.org/https://doi.org/10.1016/j.snb.2020.129194)
- 25 Zhou, S. *et al.* Sub-10 parts per billion detection of hydrogen with floating gate transistors built on semiconducting carbon nanotube film. *Carbon* **180**, 41-47 (2021).
[https://doi.org:https://doi.org/10.1016/j.carbon.2021.04.076](https://doi.org/https://doi.org/10.1016/j.carbon.2021.04.076)
- 26 Darabpour, M. & Doroodmand, M. M. Fabrication of a Glow Discharge Plasma-Based Ionization Gas Sensor Using Multiwalled Carbon Nanotubes for Specific Detection of Hydrogen at Parts Per Billion Levels. *IEEE Sensors Journal* **15**, 2391-2398 (2015).
[https://doi.org:10.1109/JSEN.2014.2369738](https://doi.org/10.1109/JSEN.2014.2369738)
- 27 Lee, J. S., Kim, S. G., Cho, S. & Jang, J. Porous palladium coated conducting polymer nanoparticles for ultrasensitive hydrogen sensors. *Nanoscale* **7**, 20665-20673 (2015).
[https://doi.org:10.1039/C5NR06193H](https://doi.org/10.1039/C5NR06193H)
- 28 Shin, D. H. *et al.* Flower-like Palladium Nanoclusters Decorated Graphene Electrodes for Ultrasensitive and Flexible Hydrogen Gas Sensing. *Scientific Reports* **5**, 12294 (2015).
[https://doi.org:10.1038/srep12294](https://doi.org/10.1038/srep12294)
- 29 Kim, K.-H. *et al.* Long-term reliable wireless H₂ gas sensor via repeatable thermal refreshing of palladium nanowire. *Nature Communications* **15** (2024). [https://doi.org:10.1038/s41467-024-53080-0](https://doi.org/10.1038/s41467-024-53080-0)
- 30 Harumoto, T., Song, J., Lin, Y.-H. & Shi, J. Nanometer-Thick Palladium–Cobalt Alloy Films for Hydrogen Sensors and Hydrogen-Mediated Devices. *ACS Applied Nano Materials* (2025).
[https://doi.org:10.1021/acsnm.5c00392](https://doi.org/10.1021/acsnm.5c00392)
- 31 Luong, H. M. *et al.* Sub-second and ppm-level optical sensing of hydrogen using templated control of nano-hydride geometry and composition. *Nature Communications* **12**, 2414 (2021).
[https://doi.org:10.1038/s41467-021-22697-w](https://doi.org/10.1038/s41467-021-22697-w)
- 32 Park, B., Na, S. Y. & Bae, I.-G. Uniform two-dimensional crystals of polystyrene nanospheres fabricated by a surfactant-assisted spin-coating method with polyoxyethylene tridecyl ether. *Scientific Reports* **9** (2019). [https://doi.org:10.1038/s41598-019-47990-z](https://doi.org/10.1038/s41598-019-47990-z)
- 33 Lulek, E. & Ertas, Y. N. Simple and Rapid Monolayer Self-Assembly of Nanoparticles at the Air/Water Interface. *Langmuir* **40**, 18039-18048 (2024).
[https://doi.org:10.1021/acs.langmuir.4c01622](https://doi.org/10.1021/acs.langmuir.4c01622)
- 34 Chen, C. *et al.* Screen-Printing Technology for Scale Manufacturing of Perovskite Solar Cells. *Advanced Science* **10** (2023). [https://doi.org:10.1002/advs.202303992](https://doi.org/10.1002/advs.202303992)
- 35 Jo, M.-S. *et al.* Ultrafast (~0.6 s), Robust, and Highly Linear Hydrogen Detection up to 10% Using Fully Suspended Pure Pd Nanowire. *ACS Nano* **17**, 23649-23658 (2023).
[https://doi.org:10.1021/acsnano.3c06806](https://doi.org/10.1021/acsnano.3c06806)

- 36 Nguyen, T. D., Wang, F., Li, X.-G., Ehrenfreund, E. & Vardeny, Z. V. Spin diffusion in fullerene-based devices: Morphology effect. *Physical Review B* **87** (2013). <https://doi.org/10.1103/physrevb.87.075205>
- 37 Samad, B. A., Belanger, É. & Duguay, C. The effect of substrate deposition temperature on the electrical and optical properties of C60 thin films. *Journal of Applied Physics* **132** (2022). <https://doi.org/10.1063/5.0099291>
- 38 Morgan, C., Schmalbuch, K., García-Sánchez, F., Schneider, C. M. & Meyer, C. Structure and magnetization in CoPd thin films and nanocontacts. *Journal of Magnetism and Magnetic Materials* **325**, 112-116 (2013). <https://doi.org/10.1016/j.jmmm.2012.07.052>
- 39 Östergren, I. *et al.* Highly Permeable Fluorinated Polymer Nanocomposites for Plasmonic Hydrogen Sensing. *ACS Applied Materials & Interfaces* **13**, 21724-21732 (2021). <https://doi.org/10.1021/acsami.1c01968>
- 40 Anděra, V. & Bastl, Z. XPS and XAES study of the interaction of palladium and copper overlayers with fullerene films. *Czechoslovak Journal of Physics* **43**, 863-868 (1993). <https://doi.org/10.1007/bf01595270>
- 41 Ngene, P. *et al.* Polymer-Induced Surface Modifications of Pd-based Thin Films Leading to Improved Kinetics in Hydrogen Sensing and Energy Storage Applications. *Angewandte Chemie International Edition* **53**, 12081-12085 (2014). <https://doi.org/10.1002/anie.201406911>
- 42 Yun, S. & Ted Oyama, S. Correlations in palladium membranes for hydrogen separation: A review. *Journal of Membrane Science* **375**, 28-45 (2011). <https://doi.org/10.1016/j.memsci.2011.03.057>
- 43 Lv, R., Zhang, Q., Wang, W., Lin, Y. & Zhang, S. ZnO@ZIF-8 Core-Shell Structure Gas Sensors with Excellent Selectivity to H₂. *Sensors* **21**, 4069 (2021). <https://doi.org/10.3390/s21124069>
- 44 Xu, T. *et al.* Self-assembled monolayer-enhanced hydrogen sensing with ultrathin palladium films. *Applied Physics Letters* **86**, 203104 (2005). <https://doi.org/10.1063/1.1929075>
- 45 Favier, F. D. R., Walter, E. C., Zach, M. P., Benter, T. & Penner, R. M. Hydrogen Sensors and Switches from Electrodeposited Palladium Mesowire Arrays. *Science* **293**, 2227-2231 (2001). <https://doi.org/10.1126/science.1063189>
- 46 Weckesser, J. *et al.* Binding and ordering of C60 on Pd(110): Investigations at the local and mesoscopic scale. *The Journal of Chemical Physics* **115**, 9001-9009 (2001). <https://doi.org/10.1063/1.1410391>
- 47 Kaushik, S., Khanderao, A. G., Gupta, P., Raghavendra Reddy, V. & Kumar, D. Growth of ultrathin Cobalt on fullerene (C60) thin-film: in-situ investigation under UHV conditions. *Materials Science and Engineering: B* **284**, 115911 (2022). <https://doi.org/10.1016/j.mseb.2022.115911>
- 48 Bing, W., Haiqian, W., Yongqing, L., Bin, X. & Jianguo, H. Structure and conductivity of C60/Pd multilayer films. *Physica C: Superconductivity* **282-287**, 735-736 (1997). [https://doi.org/10.1016/S0921-4534\(97\)00553-4](https://doi.org/10.1016/S0921-4534(97)00553-4)
- 49 Lee, E., Lee, J. M., Koo, J. H., Lee, W. & Lee, T. Hysteresis behavior of electrical resistance in Pd thin films during the process of absorption and desorption of hydrogen gas. *International Journal of Hydrogen Energy* **35**, 6984-6991 (2010). <https://doi.org/10.1016/j.ijhydene.2010.04.051>
- 50 Nugroho, F. A. A. *et al.* Inverse designed plasmonic metasurface with parts per billion optical hydrogen detection. *Nature Communications* **13**, 5737 (2022). <https://doi.org/10.1038/s41467-022-33466-8>
- 51 Tomeček, D. *et al.* Neural network enabled nanoplasmonic hydrogen sensors with 100 ppm limit of detection in humid air. *Nature Communications* **15**, 1208 (2024). <https://doi.org/10.1038/s41467-024-45484-9>

- 52 Lin, X. *et al.* Unlocking Predictive Capability and Enhancing Sensing Performances of
Plasmonic Hydrogen Sensors via Phase Space Reconstruction and Convolutional Neural
53 Networks. *ACS Sensors* **9**, 3877-3888 (2024). <https://doi.org:10.1021/acssensors.3c02651>
- 54 Simona H. Murph, K. J. L., Henry T. Sessions, Michael A. Brown. Controlled release of
hydrogen from composite nanoparticles. US patent US-10507452-B2 (2019).
- 55 Ai, B. & Zhao, Y. Glancing angle deposition meets colloidal lithography: a new evolution in the
design of nanostructures. **8**, 1-26 (2019). <https://doi.org:doi:10.1515/nanoph-2018-0105>
- 56 Hawkeye, M. M. & Brett, M. J. Glancing angle deposition: Fabrication, properties, and
applications of micro- and nanostructured thin films. *Journal of Vacuum Science & Technology*
A **25**, 1317-1335 (2007). <https://doi.org:10.1116/1.2764082>
- 57 Schimo, G., Burgstaller, W. & Hassel, A. W. Influence of atmospheric oxygen on hydrogen
detection on Pd using Kelvin probe technique. *Journal of Solid State Electrochemistry* **22**, 495-
504 (2018). <https://doi.org:10.1007/s10008-017-3715-z>
- 58 Gaaz, T. S. *et al.* Properties and Applications of Polyvinyl Alcohol, Halloysite Nanotubes and
Their Nanocomposites. *Molecules* **20**, 22833-22847 (2015).
- Yang, K. *et al.* Array-Assisted SERS Microfluidic Chips for Highly Sensitive and Multiplex
Gas Sensing. *ACS Applied Materials & Interfaces* **12**, 1395-1403 (2020).
<https://doi.org:10.1021/acami.9b19358>

Response Letter 2:

We sincerely thank all the reviewers for their time and insightful comments that helped further improve the manuscript substantially. Below are our point-by-point responses to reviewers' comments.

Responses to Comments from Reviewer #1

Authors already addressed properly

Authors' response: We sincerely thank the reviewer for their positive evaluation and are glad that our previous revision has addressed the concerns satisfactorily.

Responses to Comments from Reviewer #2

First I would like to thank the Authors for doing an extraordinary work for their revision. As such, I think now this is an exceptional work and therefore I recommend for publication. I believe this will be accepted very well by the community and will be an important piece for the future development. However if I may request one more thing? There are a few valuable discussion in the Response Letter that does not end up in the final manuscript, for example, comment 1.7. I think it will be a "waste" if it just end there, and there is very likely that some readers would have similar question/concern. Maybe authors can somewhat include this in the SI?

Anyway, thanks again and congratulations!!

Authors' response: We sincerely thank the reviewer for their encouraging and thoughtful feedback. We greatly appreciate the recognition of our efforts and are pleased to hear that the work is considered valuable for the community.

In response to the suggestion, we agree that some of the discussions provided in the response letter offer useful insights for readers. To ensure these points are not lost, we have incorporated the relevant discussion into Supplementary Information (SI) as suggested. Thank you again for your support and constructive input.

Changes made to the manuscript:

- The discussion in **Comment 1.7** of the first response letter regarding the scalability, feasibility, and potential challenges related to the mass production and integration of our PdCo hexagonal nano-network sensors has been added to the **Supplementary Note 10**. The following paragraph has been added to the manuscript (**page 27, from line 16**):

“It is noteworthy that the high-performance sensors were fabricated using simple, scalable methods that are inherently compatible with standard semiconductor processes, making the design a strong candidate for industrial integration. A detailed discussion of the fabrication steps, scalability, and potential challenges related to large-area uniformity, material cost, integration, and long-term stability is provided in Supplementary Note 10.”

- The discussion in **Comment 2.5** of the first response letter regarding the need for ppb-LOD sensors has been added to the Introduction of the manuscript (**page 4, from line 10**):

“Achieving ppb or sub-ppb detection limits not only benchmarks sensor performance under noise-limited conditions requiring high signal-to-noise ratios and advanced processing,^{1,2} but also enables critical applications such as trace hydrogen detection in deep-space missions,³ the monitoring of low-abundance hydrogen isotopes like deuterium and tritium in nuclear environments,⁴ and breath tests for assessing gastrointestinal health.^{5,6}”

References have been added to the manuscript:

- 17 Lin, X. *et al.* Unlocking Predictive Capability and Enhancing Sensing Performances of Plasmonic Hydrogen Sensors via Phase Space Reconstruction and Convolutional Neural Networks. *ACS Sensors* **9**, 3877-3888 (2024). <https://doi.org:10.1021/acssensors.3c02651>
- 18 Tian, J. *et al.* A Ppb-level hydrogen sensor based on activated Pd nanoparticles loaded on oxidized nickel foam. *Sensors and Actuators B: Chemical* **329**, 129194 (2021). <https://doi.org:https://doi.org/10.1016/j.snb.2020.129194>
- 19 Simona H. Murph, K. J. L., Henry T. Sessions, Michael A. Brown. Controlled release of hydrogen from composite nanoparticles. US patent US-10507452-B2 (2019).
- 20 Shin, W. Medical applications of breath hydrogen measurements. *Analytical and Bioanalytical Chemistry* **406**, 3931-3939 (2014). <https://doi.org:10.1007/s00216-013-7606-6>
- 21 Erdrich, S., Tan, E. C. K., Hawrelak, J. A., Myers, S. P. & Harnett, J. E. Hydrogen–methane breath testing results influenced by oral hygiene. *Scientific Reports* **11** (2021). <https://doi.org:10.1038/s41598-020-79554-x>

- **Figure R5** in **Comment 3.3** of the first response letter regarding the temperature stability of the nano-resistor network has been added as **Supplementary Figure 33**. The following paragraph has been added to the manuscript (**page 25, from line 13**):

“To address the temperature stability of the system and the temperature dependence of resistance, we evaluated sensor performance of one of our CHAs over a range from room temperature to 115 °C and consistently observed hydrogen response (see Supplementary Fig. 33). However, two notable effects emerged at elevated temperatures: (i) increased baseline drift, which may result from thermally induced morphological changes in the nanoparticle network such as expansion or coalescence that alter percolation pathways, and (ii) a significant drop in sensitivity, consistent with suppressed β -phase hydride formation and enhanced hydrogen desorption kinetics.⁷ These findings indicate that room temperature offers the best balance between baseline stability and sensing performance for our PdCo nano-network structure.”

Responses to Comments from Reviewer #3

In their response, the authors have adequately answered the questions, and the manuscript and supplementary materials have been carefully revised. While most of the responses from the authors were well understood, the answers raised additional questions. Therefore, we consider the manuscript acceptable after addressing the following comments.

Authors' response: We sincerely thank the reviewer for their positive evaluation of our first revision. We are happy to have an opportunity to further address the reviewer's critical remarks and important questions.

Can nitrogen molecules penetrate the C60 layer? If the authors claim that measurements in vacuum mode allow evaluation of the intrinsic properties of the sensing material (C60/PdCo/TAF), they should mention the diffusion of N₂ and H₂ gases in C60.

If N₂ molecules can penetrate the C60 layer, the diffusion rate of hydrogen molecules in the C60 layer in flow mode may be slower than that in vacuum mode because of the existence of interdiffusion of N₂ and H₂ in the C60 layer. In other words, the diffusion of hydrogen in the C60 layer may be the rate-limiting step for the sensing material. This behavior is intrinsic to the sensing material and is distinct from sensing system effects such as chamber geometry and flow rate. Therefore, we believe that the intrinsic response speed of the sensing material cannot be evaluated in vacuum mode. On the other hand, if the C60 layer is impermeable to N₂ gas, the C60 could act as an H₂ filter and the intrinsic reaction rate of the sensing material could be measured in vacuum mode.

Authors' response: We thank the reviewer for raising and explicitly clarifying these important questions.

First, regarding the permeability of N₂ in C₆₀, we honestly cannot determine its permeability using the sensing response since N₂ does not induce a measurable change in the signal (**Supplementary Fig. 22**). However, in one experiment using synthetic air with a thin-film sensor coated with 100 nm of C₆₀, we observed that the sensor baseline was distorted in the presence of O₂, whereas no such effect occurred in an N₂ environment (**Figure R14**). This indicates that O₂ exhibits significant permeability through the C₆₀ layer. Given that the kinetic diameter of O₂ (~0.346 nm) is comparable to that of N₂ (~0.364 nm),¹⁰ we infer that the voids in the C₆₀ lattice are also permeable to N₂. In literature, although direct measurements on pure C₆₀ films are limited, Karachevtsev *et al.* (2007) reported that both pristine and photopolymerized C₆₀ films on a polycarbonatesyloxane (PCS) allow N₂ permeation, though reduced compared to bare PCS.⁸ The N₂ permeability of the C₆₀/PCS membrane is on the order of 10⁻⁵ cm³/(cm² cbar), which corresponds to ~10⁵ Barrer.⁸ To put this number in perspective, Teflon is known for its high permeability to H₂ and its H₂ permeability is about 10³ Barrer, as reported by Östergren *et al.* (2021).⁹ These together suggest that C₆₀ is relatively high permeable to N₂.

Figure R14. Sensing responses of 15 nm PdCo thin film with 100 nm C_{60} top coating sensor to 2% H_2 in N_2 and in synthetic air measured in flow mode with a total flow rate of 400 sccm.

Second, if N_2 diffuses through C_{60} , its interdiffusion with H_2 could hinder H_2 transport due to molecular collisions inside the C_{60} polycrystal. This effect is expected to be more pronounced at low H_2 partial pressures, where the relative influence of N_2 is greater. Nonetheless, the permeation inhibition by N_2 is not unique to C_{60} . It also happens in uncoated Pd membranes due to concentration polarization where N_2 accumulates near the surface and reduces the effective driving force for H_2 permeation.¹¹ It is worth noting that in our design, the C_{60} layer is positioned underneath the sensing stack, while the top surface remains Pd (coated with Teflon) directly exposed to the gas stream. To study the effect of N_2 on H_2 diffusion kinetics, we designed an additional experiment with one of CHA sensors (20 nm $C_{60}/5$ nm PdCo CHA₃₀₀) and measured its response under two different gas environments: pure H_2 and H_2/N_2 mixture (4% H_2 in N_2) in vacuum mode (**Figure R15**). The sensitivity and response time t_{90} are nearly identical within experimental errors across H_2 partial pressure range of 0.1 – 10 mbar (corresponding to 100 ppm – 1 vol.% H_2) between the two gas sources, indicating consistent sensor performance regardless of the presence of N_2 . These results suggest that the concentration polarization of H_2/N_2 mixture has a negligible effect on the sensing performance under vacuum mode measurement within the tested P_{H_2} range.

Figure R15. **a** Sensitivity $\Delta R/R$ and **b** absorption time t_{90} of 20 nm $C_{60}/5$ nm PdCo CHA₃₀₀ sensor measured in vacuum mode using pure H_2 and 4% H_2 in N_2 balance as the gas sources.

Last but not least, vacuum mode measurements eliminate external flow effects and allow clearer evaluation of internal transport through the layered structure. While some interdiffusion may still occur within the C₆₀ layer, vacuum-mode testing helps isolate intrinsic diffusion behavior and may clarify the influence of mixed versus pure gas feeds.

Change made to the manuscript:

- **Figure R15** has been added to the SI as **Supplementary Figure 23**.
- **Supplementary Note 3** (**page S11, from line 6**) has been added revised accordingly where “the sensing element’s intrinsic kinetics” is replaced by “the internal transport through layered structures”:

“In contrast, vacuum mode eliminates these transport limitations as the gas pressure reaches equilibrium in < 100 ms (Supplementary Fig. 4) and allows us to isolate **the internal transport through the layered structure.**”

- The following paragraph has been added to the manuscript (**page 20, from line 6**):

“Although C₆₀ is reported to be high permeable to N₂,⁸ the sensor’s signal remains unaffected by the N₂ pressure variation (see **Supplementary Fig. 22a**) and the interdiffusion of N₂ with H₂ has a negligible effect on the sensing performance under vacuum mode measurement within the tested P_{H₂} range as illustrated in **Supplementary Fig. 23**. Thus, H₂ and N₂ mixture can be utilized in vacuum mode measurement to reliably extract the sensitivity of the sensor at low partial H₂ pressures.”

References have been added to the manuscript:

- 54 Karachevtsev, V. A. *et al.* Permeability of C₆₀ films deposited on polycarbonatesyloxane to N₂, O₂, CH₄, and He gases. *Applied Surface Science* **253**, 3062-3065 (2007).
<https://doi.org/https://doi.org/10.1016/j.apsusc.2006.06.053>

Rebuttal References 2

- 1 Tomeček, D. *et al.* Neural network enabled nanoplasmonic hydrogen sensors with 100 ppm limit of detection in humid air. *Nature Communications* **15**, 1208 (2024).
<https://doi.org/10.1038/s41467-024-45484-9>
- 2 Lin, X. *et al.* Unlocking Predictive Capability and Enhancing Sensing Performances of Plasmonic Hydrogen Sensors via Phase Space Reconstruction and Convolutional Neural Networks. *ACS Sensors* **9**, 3877-3888 (2024). <https://doi.org/10.1021/acssensors.3c02651>
- 3 Tian, J. *et al.* A Ppb-level hydrogen sensor based on activated Pd nanoparticles loaded on oxidized nickel foam. *Sensors and Actuators B: Chemical* **329**, 129194 (2021).
[https://doi.org:https://doi.org/10.1016/j.snb.2020.129194](https://doi.org/https://doi.org/10.1016/j.snb.2020.129194)
- 4 Simona H. Murph, K. J. L., Henry T. Sessions, Michael A. Brown. Controlled release of hydrogen from composite nanoparticles. US patent US-10507452-B2 (2019).
- 5 Shin, W. Medical applications of breath hydrogen measurements. *Analytical and Bioanalytical Chemistry* **406**, 3931-3939 (2014). <https://doi.org/10.1007/s00216-013-7606-6>
- 6 Erdrich, S., Tan, E. C. K., Hawrelak, J. A., Myers, S. P. & Harnett, J. E. Hydrogen–methane breath testing results influenced by oral hygiene. *Scientific Reports* **11** (2021).
<https://doi.org/10.1038/s41598-020-79554-x>
- 7 Darmadi, I., Nugroho, F. A. A. & Langhammer, C. High-Performance Nanostructured Palladium-Based Hydrogen Sensors—Current Limitations and Strategies for Their Mitigation. *ACS Sensors* **5**, 3306-3327 (2020). <https://doi.org/10.1021/acssensors.0c02019>
- 8 Karachevtsev, V. A. *et al.* Permeability of C60 films deposited on polycarbonatesyloxane to N₂, O₂, CH₄, and He gases. *Applied Surface Science* **253**, 3062-3065 (2007).
[https://doi.org:https://doi.org/10.1016/j.apsusc.2006.06.053](https://doi.org/https://doi.org/10.1016/j.apsusc.2006.06.053)
- 9 Östergren, I. *et al.* Highly Permeable Fluorinated Polymer Nanocomposites for Plasmonic Hydrogen Sensing. *ACS Applied Materials & Interfaces* **13**, 21724-21732 (2021).
<https://doi.org/10.1021/acami.1c01968>
- 10 Wikipedia contributors, "Kinetic diameter," *Wikipedia, The Free Encyclopedia*, https://en.wikipedia.org/w/index.php?title=Kinetic_diameter&oldid=1209540273 (accessed July 28, 2025).
- 11 Caravella, A., Barbieri, G. & Drioli, E. Concentration polarization analysis in self-supported Pd-based membranes. *Separation and Purification Technology* **66**, 613-624 (2009).
[https://doi.org:https://doi.org/10.1016/j.seppur.2009.01.008](https://doi.org/https://doi.org/10.1016/j.seppur.2009.01.008)

Response Letter 3:

We sincerely thank the reviewers for their time and insightful comments that helped further improve the manuscript substantially. Below are our point-by-point responses to reviewers' comments.

Responses to Comments from Reviewer #3

I have understood that the existence of nitrogen hardly affected the reaction time as shown in figure R15 and the nitrogen gas diffusion in C60 layer is not important to evaluate the intrinsic material sensitivity. The authors have addressed my question appropriately. I recommend that the revised manuscript is published to nature communications.

Authors' response: We sincerely thank the reviewer for their positive evaluation and are glad that our previous revision has addressed the concerns satisfactorily.